# SYMPHONY: Synergistic Multi-agent Planning with Heterogeneous Language Model Assembly

**Wei Zhu**     **Zhiwen Tang**[*]   **Kun Yue**

School of Information Science and Engineering, Yunnan University, Kunming, China

Yunnan Key Laboratory of Intelligent Systems and Computing, Kunming, China

`zhuwei@stu.ynu.edu.cn, {zhiwen.tang, kyue}@ynu.edu.cn`

## Abstract

Recent advancements have increasingly focused on leveraging large language models (LLMs) to construct autonomous agents for complex problem-solving tasks. However, existing approaches predominantly employ a single-agent framework to generate search branches and estimate rewards during Monte Carlo Tree Search (MCTS) planning. This single-agent paradigm inherently limits exploration capabilities, often resulting in insufficient diversity among generated branches and suboptimal planning performance. To overcome these limitations, we propose **SY**nergistic **M**ulti-agent **P**lanning with **H**eter**O**geneous la**N**gauge model assembl**Y** (**SYMPHONY** [2]), a novel multi-agent planning framework that integrates a pool of heterogeneous language model-based agents. By leveraging diverse reasoning patterns across agents, SYMPHONY enhances rollout diversity and facilitates more effective exploration. Empirical results across multiple benchmark tasks show that SYMPHONY achieves strong performance even when instantiated with open-source LLMs deployable on consumer-grade hardware. When enhanced with cloud-based LLMs accessible via API, SYMPHONY demonstrates further improvements, outperforming existing state-of-the-art baselines and underscoring the effectiveness of heterogeneous multi-agent coordination in planning tasks.

## 1 Introduction

The advent of large language models (LLMs) has significantly advanced the development of autonomous agents capable of performing complex tasks across various domains, including question answering, code generation, and web navigation. These LLM-based agents leverage the extensive knowledge and reasoning capabilities inherent in LLMs to make decisions and plan actions. A prevalent approach in this context is the integration of Monte Carlo Tree Search (MCTS) [10] with LLMs, wherein the LLM guides the exploration of potential action sequences to achieve specific goals [14, 46, 28, 12]. This combination has shown promise in enhancing the decision-making processes of autonomous agents.

Despite the recent progress in integrating LLMs with planning algorithms, existing methods [34, 33, 43, 29, 42, 14] predominantly adopt a single-model paradigm in which one LLM is queried multiple times with identical or slightly perturbed prompts to simulate diverse action branches during MCTS. The underlying assumption is that the stochasticity or sampling variance of the model is sufficient to generate rollouts that explore a wide range of potential solutions. However, in practice, this approach suffers from a critical limitation: the outputs tend to exhibit high similarity across calls, often reflecting the same dominant reasoning pattern learned by the model [14, 46, 12]. As a result, the generated rollouts lack meaningful diversity, leading to narrow and redundant search trajectories.

---

[*]Corresponding author

[2]Code is available at `https://github.com/ZHUWEI-hub/SYMPHONY`

39th Conference on Neural Information Processing Systems (NeurIPS 2025).

This deficiency severely constrains the agent's exploration capability within the solution space. When the search tree is populated with highly similar branches, the planner becomes susceptible to local optima, and its ability to discover novel or unexpected solutions is greatly diminished. In particularly challenging tasks that require compositional reasoning or multi-step tool use, the agent may fail to identify the correct solution path altogether. Even in cases where the solution is eventually found, the process may involve excessive sampling and token consumption, incurring significant computational overhead. These inefficiencies highlight a fundamental mismatch between the need for diverse exploration in planning and the limited variability achievable by repeatedly sampling from a single, monolithic LLM.

To address the above limitations, we propose **SY**nergistic **M**ulti-agent **P**lanning with **H**eterogene**O**us La**N**guage Model Assembl**Y** (**SYMPHONY**). The framework integrates multiple language models into a unified planning system that enhances multi-step reasoning through diversity-aware search, adaptive coordination, and reflective adaptation.

A central innovation of SYMPHONY is its heterogeneous agent pool, composed of LLMs with diverse pretraining sources and reasoning styles. Instead of relying on a single agent, SYMPHONY assigns different agents to generate candidate actions at each search node, thereby introducing structural diversity into the search tree. This diversity increases the likelihood of generating complementary reasoning paths, reduces model-specific biases, and improves performance on complex, multi-hop tasks. Empirical results show that expanding model diversity leads to more unique branches per node and consistent gains in task accuracy.

In addition to model heterogeneity, SYMPHONY incorporates several complementary components that further enhance planning performance. A UCB-based scheduling strategy dynamically allocates agents based on historical effectiveness, improving coordination across agents. An entropy-modulated confidence scoring (EMCS) mechanism calibrates value estimates using agent-level uncertainty, yielding more stable evaluations. Finally, a pool-wise memory sharing mechanism enables agents to learn from past failures through natural language reflections, which are shared across the agent pool and incorporated into future prompts. These components together support efficient, adaptive, and robust search behavior.

We evaluate SYMPHONY across three distinct environments that represent key capabilities of LLM-based agents: multi-hop reasoning (HotpotQA), sequential decision making (WebShop), and code generation (MBPP). Experimental results show that SYMPHONY consistently outperforms strong baselines. In addition to improved performance, SYMPHONY achieves higher planning efficiency, requiring fewer MCTS node expansions to reach correct solutions. Notably, the framework delivers competitive or superior results even when built upon cost-effective models, demonstrating practical value without relying on high-cost large-scale deployments.

## 2 Related Work

### 2.1 LLM-based Planning and Reasoning

Early work on LLM-based reasoning focused on improving consistency and correctness through guided inference. Brown et al. [5] introduced in-context learning with exemplars, while the Chain-of-Thought paradigm [34, 19, 24] encouraged models to generate step-by-step rationales during prediction. Later studies proposed more structured prompting techniques, such as meta-prompting [45] and meta-constraint-guided inference [30], to scaffold the reasoning process with predefined formats or global constraints.

To enhance reasoning adaptability, several methods incorporate dynamic feedback. Yao et al. [43] interleaves environment interactions with reasoning steps, Shinn et al. [29] enables self-correction through natural language reflections, and code-based approaches Qiu et al. [27], Chen et al. [9] iteratively refine outputs based on execution results. Xu et al. [37] further improves performance by prompting LLMs to rearticulate and revise their own reasoning chains.

As tasks grew more complex, researchers began to move beyond linear inference and explore tree-structured reasoning. Tang et al. [31] and Zhang et al. [44] introduced early mechanisms for maintaining and refining multiple hypotheses in dialogue and QA tasks. Wang et al. [32] aggregates diverse reasoning paths through sampling and majority voting, while Pan et al. [25] dynamically adjusts decoding strategy between heuristic and deliberative modes.

A more principled formulation of structured search appears in Tree of Thoughts [42], which organizes reasoning steps into a decision tree with intermediate backtracking. Building on this, Monte Carlo Tree Search (MCTS) has been applied to guide reasoning more systematically: Hao et al. [14] treat the LLM as a world model within a reward-driven search process, and Zhou et al. [46] further integrate reasoning, planning, and reflection within an MCTS-based framework using learned value estimates and external feedback. Shi et al. [28] employ memory-augmented single-agent MCTS to enhance decision-making in text-based games.

While prior work focuses on structured reasoning with a single model, our approach introduces model-level heterogeneity to enhance diversity and robustness in planning.

## 2.2 Multi-Agent Collaboration with Language Models

Multi-agent frameworks leverage multiple LLMs or specialized modules to improve reasoning diversity, adaptability, and robustness. Early approaches adopt static task division, assigning agents predefined roles and communication protocols. For instance, ChatDev [26] simulates software development by dividing planning, coding, and testing among fixed-role agents, while MetaGPT [15] enforces similar pipelines using hand-crafted coordination logic. AutoAgents [7] automates agent instantiation but still operates under rigid, rule-based interaction patterns. Although effective in structured environments, these systems struggle with dynamic tasks due to their limited flexibility.

More recent work shifts toward dynamic coordination, enabling emergent collaboration and context-aware adaptation. AgentVerse [8] adopts a blackboard architecture where agents communicate freely through shared language-based memory. CAMEL [20] introduces turn-based agent dialogue for zero-shot task-solving, while AutoGen [35] allows agents to negotiate roles and delegate subtasks on the fly. In complex QA settings, WebGPT [23] decomposes queries into search, summarization, and synthesis subtasks, and MAd [21] employs adversarial debate between LLMs to expose reasoning flaws. MASTER [12] integrates multi-agent behavior into MCTS by adapting the UCT formula using reward signals. AgentCoder [16] further demonstrates the utility of functional specialization in code generation, coordinating programmer, test designer, and test executor agents within a feedback loop to ensure correctness and completeness.

Unlike existing frameworks that rely on uniform agents and costly coordination, our method enables lightweight, heterogeneous collaboration through principled search and memory sharing.

# 3 SYMPHONY

## 3.1 Background and Overview

A Markov Decision Process (MDP) [4] provides a principled framework for modeling sequential decision-making, defined by the tuple $(S, A, \mathcal{T}, R, \gamma)$, where $S$ is the state space, $A$ is the action space, $\mathcal{T} : S \times A \to \mathcal{P}(S)$ defines the transition dynamics, $R : S \times A \to \mathbb{R}$ is the reward function, and $\gamma \in [0, 1]$ is the discount factor. At each timestep, the agent observes a state $s \in S$, selects an action $a \in A$, transitions to a new state $s' \sim P(\cdot \mid s, a)$, and receives a reward $R(s, a)$. The objective is to learn a policy that maximizes the expected cumulative discounted return.

LLMs can be naturally integrated into this framework to support high-level reasoning and decision-making. Specifically, an LLM can serve as a *policy* by generating actions conditioned on language-based state representations, as a *value function* by estimating expected returns from textual trajectories, or as a *world model* by predicting future states and rewards through learned knowledge. Unlike traditional reinforcement learning agents that rely on explicit environment modeling and manually designed reward signals, LLM-based agents leverage pretraining on large corpora to internalize commonsense, domain knowledge, and structured reasoning. This allows them to operate effectively in complex, open-ended environments with minimal task-specific engineering.

Monte Carlo Tree Search (MCTS) [10] is a sample-based planning algorithm that incrementally builds a search tree by balancing exploration and exploitation. It has been widely used in sequential decision-making problems and is well-suited for integration with LLM-based agents, as it allows structured reasoning guided by model-generated priors.

Formally, given the MDP setup, MCTS constructs a partial search tree rooted at the initial state $s_0$, iteratively performing four steps: *selection*, which traverses the tree using an upper confidence

bound to choose promising actions; *expansion*, which adds new child nodes for unexplored actions; *simulation* (or rollout), which estimates future rewards using a policy; and *backpropagation*, which updates statistics along the visited path. A detailed description of MCTS can be found in Appendix C.

In this work, we adapt MCTS by incorporating LLMs to guide both the selection and rollout phases, replacing uniform or heuristic strategies with model-informed priors that focus exploration on semantically meaningful regions. Building on this foundation, we introduce **SYMPHONY**, a synergistic multi-agent planning framework designed to enhance both the efficiency and robustness of LLM-based decision-making. SYMPHONY extends classical MCTS through several key innovations: a heterogeneous ensemble of LLM agents with diverse inductive priors, a UCB-driven adaptive agent scheduling strategy, a pool-wise memory sharing protocol enabling decentralized reflective adaptation, and an entropy-aware utility modulation mechanism for confidence-calibrated evaluation. These components collectively promote diverse trajectory generation, context-aware coordination, coherent information propagation and reliable value estimation. The theoretical analysis and complete pseudocode of SYMPHONY can be found in Appendix A.

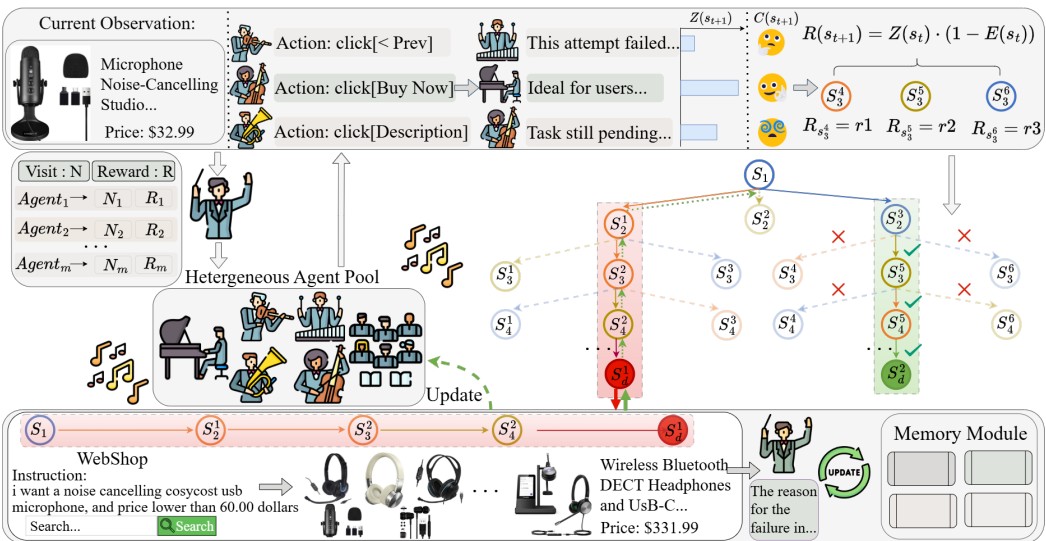

Figure 1: SYMPHONY System Overview.

## 3.2 Heterogeneous Agent Pool

The heterogeneous agent pool in SYMPHONY is designed to enhance rollout diversity by incorporating multiple language models with varied inductive biases and reasoning behaviors. Unlike traditional MCTS approaches that rely on repeated queries to a single language model, SYMPHONY maintains a collection of distinct language models, each serving as an independent agent that contribute complementary perspectives during search. Formally, the agent pool is represented as $\mathcal{M}^{(k)} = \left\{ M_1^{(k)}, \cdots, M_n^{(k)} \right\}$, where $M_i$ is the $i^{th}$ agent based on a language model after the $k^{th}$ memory update.

These agents may be instantiated from either open-source models that are deployable on consumer-grade hardware, or large-scale cloud-based models accessible only via remote API. Different agents exhibit complementary strengths in reasoning depth, factual precision, abstraction ability, and stylistic preferences, which collectively enhance the system's capacity to explore diverse trajectories in the search space.

SYMPHONY employs a uniform input-output interface for agents pool. More specifically, the input to the agent pool at the $t^{th}$ step is $P_\phi(s_t, h_{t-1})$, where $\phi \in \{\text{expansion}, \text{evaluation}, \text{reflection}\}$ is the function indicator of language models , $P_\phi$ is the corresponding prompt template, and $h_t$ is the interaction history $h_{t-1} = (s_0, a_0, \cdots, s_{t-1}, a_{t-1})$. This design choice facilitates modularity. New models can be added or removed without altering the core planning algorithm. It ensures compatibility with future advances in LLMs and facilitates efficient reuse of available computational

resources under different deployment settings. Prompts for each stage can be found in case studies (Appendix L).

### 3.3 Agent Scheduling

To operationalize the functional heterogeneity of the agent pool, SYMPHONY implements an adaptive dispatch mechanism grounded in the Upper Confidence Bound (UCB) principle, formulating agent selection at each MCTS rollout step as a structured multi-armed bandit problem. Rather than relying on static sampling heuristics or fixed priority weights, the framework dynamically calibrates agent choice based on performance statistics, enabling context-sensitive allocation of reasoning capacity.

Formally, for each agent $M_i^{(k)} \in \mathcal{M}^{(k)}$, the scheduler maintains a cumulative utility estimate $\bar{Q}(M_i^{(k)})$ reflecting empirical rollout effectiveness, Let $S_{M_i^{(k)}}$ denotes the set of nodes generated by agent $M_i^{(k)}$, $S_{M_i^{(k)}} = \{s_{t+1} \sim \mathcal{T}(s_t, M_i^{(k)}(s_t, h_{t-1}))\}$. We record the total invocation count for agent $M_i^{(k)}$ as $N_i^{(k)}$, Similarly, the cumulative average score for agent $M_i^{(k)}$ is defined as $\bar{Q}(M_i^{(k)}) = \sum_{s_t \in S_{M_i^{(k)}}} R(s_t)/|S_{M_i^{(k)}}|$. The selection priority at a search node $s_t$ is governed by the canonical UCB expression:

$$\text{UCB}(M_i^{(k)}) = \bar{Q}(M_i^{(k)}) + \alpha \cdot \sqrt{\frac{\ln N_{total}^{\mathcal{M}^{(k)}}}{N(M_i^{(k)}) + 1}} \tag{1}$$

Here $\alpha$ denotes an exploration–exploitation trade-off hyperparameter, $N_{total}^{\mathcal{M}^{(k)}} = \sum_{j=1}^{n} N(M_j^{(k)})$ represents the total number of scheduling decisions made thus far, and the denominator smoothing term ensures initialization-phase optimism. This formulation favors agents that exhibit either superior historical returns or low invocation frequency, thereby enabling simultaneous exploitation of high-confidence models and exploration of underutilized reasoning modes.

Crucially, this scheduling mechanism is not an isolated module but is tightly interwoven with the recursive structure of MCTS, encompassing action generation and reflective evaluation. Following UCT-guided node traversal, a frontier node $s_t$ is expanded by dispatching an $M_i^{(k)} \in \mathcal{M}^{(k)}$ selected via Equation 1, which is queried using the expansion prompt:

$$a_t = M_i^{(k)}(P_{\text{expansion}}(s_t, h_{t-1})) \tag{2}$$

where $h_{t-1}$ encodes the accumulated interaction trace. The returned actions populate the search frontier with semantically diverse and structurally varied hypotheses.

The agent scheduling mechanism is also used in creating pool-wise reflection memory and node evaluation with EMCS, which will be detailed in the subsequent subsections.

We further establish, from a theoretical perspective, that sampling agents from the ensemble with non-zero probabilities leads to a strictly lower expected error than deterministically selecting a single agent. The detailed proof is provided in Appendix B.

### 3.4 Pool-wise Memory Sharing

To support continual adaptation without parameter updates, SYMPHONY introduces a pool-wise memory sharing mechanism based on decentralized reflection with natural language. Rather than relying on explicit retraining, agents update their behavior by integrating peer-generated reflections into prompt-level memory.

When a trajectory terminates unsuccessfully, $\tau_{\text{fail}} = (s_0, a_0, \ldots, s_T)$, a UCB-selected agent $M_i^{(k)}$ generates a structured reflection $\mathcal{R}_i^k$ summarizing the failure. This reflection is broadcast to the entire agent pool and treated as a shared memory block. As reflections accumulate from different agents and episodes, they form a diverse collective memory that enhances generalization and coordination.

To manage memory constraints and maintain efficiency, each agent retains a fixed-size buffer updated via a FIFO policy. Reflections are incorporated through prompt-level memory updates:

$$\mathcal{M}^{(k+1)} = \text{Update}(\mathcal{M}^{(k)}, \mathcal{R}^k), \mathcal{R}^k = M_i^{(k)}(P_{\text{reflection}}(s_t, h_{t-1})) \tag{3}$$

This update mechanism enables behavioral adjustment without modifying model parameters, supporting lightweight and scalable adaptation across heterogeneous agents.

### 3.5 Entropy-Modulated Node Evaluation

To improve value estimation during search, SYMPHONY introduces an entropy-modulated node evaluation strategy that adjusts utility scores based on agent confidence. Upon expanding a new node $s_t$, a scheduled agent $M_i^{(k)} \in \mathcal{M}^{(k)}$ performs an internal evaluation, producing a value estimate $Z(s_t) \in [0, 1]$ and a confidence score $C(s_t) \in (0, 1)$:

$$Z(s_t), C(s_t) = M_i(P_{\text{evaluation}}(s_t, h_{t-1})) \tag{4}$$

To integrate these outputs, SYMPHONY employs Entropy-Modulated Confidence Scoring (EMCS), which penalizes uncertain predictions by down-weighting value estimates using the entropy of a Bernoulli distribution. Here, the confidence score $C(s_t)$ is interpreted as the success probability of a Bernoulli variable: the entropy is maximal at $C(s_t) = 0.5$, indicating maximum uncertainty, and approaches zero as $C(s_t) \to 0$ or $C(s_t) \to 1$, reflecting high confidence.

$$R(s_t) = Z(s_t) \cdot (1 - E(s_t)) \tag{5}$$

where $E(s_t) = -C(s_t) \ln C(s_t) - (1 - C(s_t)) \ln(1 - C(s_t))$.

This formulation preserves confident evaluations while suppressing uncertain ones, ensuring that nodes with ambiguous outcomes have reduced influence. Compared to fixed heuristics, EMCS offers uncertainty-aware, real-time modulation with minimal overhead, leading to more stable and reliable planning behavior within the MCTS loop.

## 4 Experiments

We evaluate our approach across three representative tasks spanning reasoning, decision-making, and code generation. Specifically, we conduct experiments on: (1) multi-hop question answering using HotpotQA [40] to assess reasoning capabilities; (2) goal-directed interaction on WebShop [41] to evaluate decision-making and planning; and (3) code generation on MBPP [3] to test the model's ability to reason and produce executable solutions.

### 4.1 Experiment Settings

SYMPHONY supports flexible agent composition and is compatible with a range of language models under different computational constraints. We evaluate two deployment configurations: **SYMPHONY-S**, designed for consumer-grade hardware, and **SYMPHONY-L**, which leverages large-scale foundation models via cloud-based APIs.

**SYMPHONY-S** comprises open-source models that can be executed locally, including Qwen2.5-7B-Instruct-1M [39], Mistral-7B-Instruct-v0.3 [18], and Llama-3.1-8B-Instruct [13]. This configuration supports efficient inference with minimal deployment cost. In contrast, **SYMPHONY-L** comprises high-performance models: GPT-4 [1], Qwen-Max (2024-09-19) [38], and DeepSeek-V3 (2025-03-24) [22], which operate through API endpoints within inference-as-a-service infrastructures.

All experiments are carried out under a unified protocol aligned with previous work [29, 46, 12]. To ensure comparability, we apply consistent prompt formats and fixed hyperparameter settings across both configurations, including decoding temperature, planning depth, rollout budget, and number of demonstrations. To mitigate LLM stochasticity, each experiment is repeated 3 times on the same data set, and the mean accuracy is reported. The detailed hyper-parameter settings are described in Appendix D.

| Table 1: HotpotQA. | |
| --- | --- |
| **Method** | **Exact Match ↑** |
| CoT [34] | 0.34 |
| CoT-SC [33] | 0.38 |
| ReAct [43] | 0.39 |
| Reflexion [29] | 0.51 |
| ToT [42] | 0.55 |
| RAP [14] | 0.60 |
| LATS [46] | 0.71 |
| Beam Retrieval [44] | 0.73 |
| MASTER [12] | 0.76 |
| **SYMPHONY-S** | **0.59** |
| **SYMPHONY-L** | **0.79** |

| Table 2: WebShop. | | |
| --- | --- | --- |
| **Method** | **Score ↑** | **SR ↑** |
| IL [41] | 0.60 | 0.29 |
| IL+RL [41] | 0.62 | 0.29 |
| ReAct [43] | 0.54 | 0.32 |
| Reflexion [29] | 0.64 | 0.35 |
| Fine-tuning [11] | 0.68 | 0.45 |
| AgentKit [36] | 0.70 | – |
| LATS [46] | 0.76 | 0.38 |
| MASTER [12] | 0.80 | – |
| Human Expert [41] | 0.82 | 0.60 |
| **SYMPHONY-S** | **0.82** | **0.56** |
| **SYMPHONY-L** | **0.88** | **0.72** |

| Table 3: MBPP. | | |
| --- | --- | --- |
| **Method** | **Pass@1 (Python) ↑** | **Pass@1 (Rust) ↑** |
| GPT-4 [29] | 0.800 | 0.710 |
| GPT-4(CoT) [12] | 0.683 | – |
| GPT-4(ReAct) [43] | 0.710 | – |
| Reflexion [29] | 0.771 | 0.754 |
| RAP [14] | 0.714 | – |
| LATS [46] | 0.811 | – |
| MetaGPT [15] | 0.877 | – |
| AgentVerse [8] | 0.890 | – |
| MASTER [12] | 0.910 | – |
| AgentCoder [17] | 0.918 | – |
| **SYMPHONY-S** | **0.927** | **0.946** |
| **SYMPHONY-L** | **0.965** | **0.974** |

Note: Metrics are normalized to the [0,1] range; A dash (–) marks those not reported in the publication.

## 4.2 Reasoning:HotpotQA

**Setup.** HotpotQA [40] is a large-scale benchmark for multi-hop question answering, constructed from Wikipedia and containing approximately 113,000 question–answer pairs. In line with prior work [43, 29, 46, 12], we employ an oracle feedback setting, where the environment immediately indicates whether a selected answer is correct. This setup is designed to isolate and evaluate the agent's decision-making capabilities during interaction, rather than its ability to generate final answers. Evaluation on this dataset is based on the exact match (EM) metric.

We compare SYMPHONY against representative baselines from four categories: (1) *Linear reasoning* methods such as CoT [34] and CoT-SC [33]; (2) *Feedback-driven* approaches including ReAct [43] and Reflexion [29]; (3) *Structured reasoning* methods such as ToT [42], RAP [14], LATS [46], and Beam Retrieval [44]; and (4) the *multi-agent framework* MASTER [12], which builds multi-agent from the same LLM. Baseline results are taken from Gan et al. [12], where GPT-4 is used uniformly across all methods.

**Results.** SYMPHONY demonstrates strong performance across all baseline categories. The lightweight **SYMPHONY-S** outperforms both linear reasoning and feedback-driven baselines, and performs comparably to structured search methods like RAP. The stronger **SYMPHONY-L** surpasses all structured baselines, including MASTER, achieving state-of-the-art performance on HotpotQA. These improvements reflect SYMPHONY's ability to combine model heterogeneity with coordinated compositional reasoning.

## 4.3 Sequential Decision Making:WebShop

**Setup.** WebShop [41] is a simulated e-commerce platform featuring over 1.18 million products and 12,000 natural language queries. Agents must navigate the website using browser-like operations (e.g., search, click, select) to identify items that satisfy user constraints. Performance is measured by average score, which reflects partial attribute satisfaction, and success rate (SR), which reflects full constraint satisfaction.

We compare SYMPHONY against a comprehensive set of baselines reflecting five categories: (1) *Task-native methods* including imitation learning (IL), IL+RL, and Human Expert [41]; (2) *Supervised models* such as a fine-tuned LLM [11]; (3) *Feedback-driven reasoning*, including ReAct, Reflexion; (4) *Structured search* methods including LATS and AgentKit; and (5) the *multi-agent framework*: MASTER [12]. All baselines were reproduced under consistent settings by Gan et al. [12] using GPT-4, ensuring fair comparison.

**Results.** SYMPHONY outperforms all baseline categories. Compared to task-native and supervised approaches, it achieves higher task completion while requiring no domain-specific training. Against feedback-driven and structured search methods, it exhibits stronger planning efficiency and generalization. Finally, SYMPHONY-L surpasses the multi-agent MASTER, establishing a new performance benchmark. These results underscore SYMPHONY's adaptability across different task-specific execution environments.

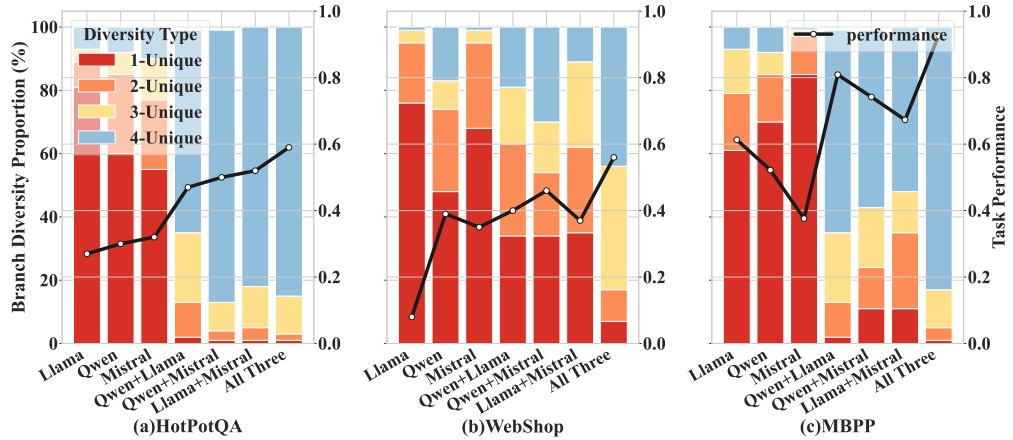

Figure 2: Branch Diversity vs. Task Performance. Bars and left y-axis shows the branch diversity, while lineplot and right y-axis shows the task performance.

## 4.4 Programming:MBPP

**Setup.** The Mostly Basic Programming Problems (MBPP) [3] involves multi-step code generation tasks that require condition decomposition, procedural planning, and implementation. Each task provides a description in natural language and a test suite. Success is defined by passing all tests. We follow [29] to evaluate both Python and Rust versions of the datasets using the MultiPL-E compiler suite [6].

Baselines span three major categories: (1) *Single-agent methods*, including GPT-4 and Reflexion [29], representing the performance ceiling of basic prompting and reactive reasoning; (2) *Multi-agent frameworks*, such as MetaGPT [15], AgentVerse [8], and AgentCoder [17], which explore different collaboration strategies; and (3) *Search-based approaches*, including RAP [14], LATS [46], and MASTER [12], which emphasize structured optimization. All baseline results are drawn from or reproduced by Gan et al. [12] under consistent backbone and data settings.

**Results.** SYMPHONY achieves strong performance across all baseline categories. Compared to single-agent methods, it demonstrates superior reasoning depth and planning efficiency. Against multi-agent frameworks, SYMPHONY provides more effective solution search via the introduction of heterogeneous agent pool. Compared to search-based approaches, it attains state-of-the-art performance in cross-language settings, including Rust, a programming language usually ignored by previous works. These results confirm SYMPHONY's robustness, generality, and computational efficiency in code generation.

## 4.5 Diversity Analysis

Branch diversity plays a crucial role in effective search. To assess its impact, we evaluate how different agent pool configurations affect both task performance and branch diversity across all three tasks using SYMPHONY-S. The expansion width is fixed at 4, and each node's candidate branches are categorized by output uniqueness: (a) **4-Unique**: all branches distinct, (b) **3-Unique**, (c) **2-Unique**, and (d) **1-Unique**: all branches identical. Higher frequencies of 3-Unique and 4-Unique indicate more diversified and informative exploration.

As shown in Figure 2, increasing agent heterogeneity, from single-agent to pairwise and full-trio configurations (e.g., Qwen+Mistral+Llama), leads to a substantial rise in 4-Unique expansions. On MBPP, for example, this proportion exceeds 80% under the full ensemble, compared to under 20% in the single-agent setting. This increase in structural diversity strongly correlates with improved accuracy, with SYMPHONY outperforming single-agent baselines by over 30% on MBPP and showing similar trends on HotpotQA and WebShop. These findings highlight the critical role of model-level diversity in enhancing search coverage and reasoning robustness.

Table 5: Ablation Study.

| Method | HotpotQA(EM)↑ | WebShop(SR)↑ | MBPP(pass@1)↑ |
|---|---|---|---|
| SYMPHONY-S | **0.59** | **0.56** | **0.927** |
| w/o Agent Scheduling | 0.51 | 0.48 | 0.906 |
| w/o Memory Sharing | 0.45 | 0.46 | 0.871 |
| w/o EMCS | 0.51 | 0.49 | 0.892 |

We also experimented with alternative diversity-promoting strategies such as adversarial prompting and temperature scaling, but found their effect to be marginal. Detailed comparisons are included in Appendix I.

## 4.6 Efficiency and Cost Analysis

Table 4: Comparison of the search tree size on HotpotQA.

| Method | K | HotpotQA ↑ | #Nodes ↓ |
|---|---|---|---|
| ToT | 10 | 0.34 | 33.97 |
| RAP | 10 | 0.44 | 31.53 |
| LATS | 10 | 0.44 | 28.42 |
| ToT | 50 | 0.49 | 84.05 |
| RAP | 50 | 0.54 | 70.60 |
| LATS | 50 | 0.61 | 66.65 |
| **SYMPHONY-S** | **10** | **0.59** | **16.39** |
| **SYMPHONY-L** | **10** | **0.79** | **9.47** |

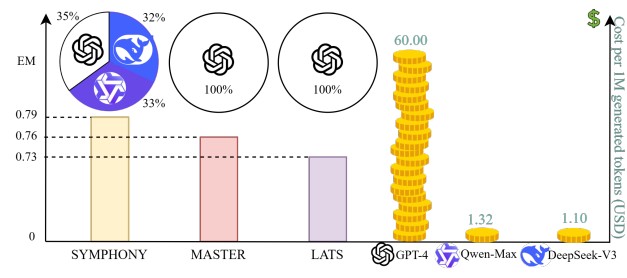

Figure 3: Comparison of model invocation frequency and final performance on HotpotQA.

To evaluate SYMPHONY's practicality, we analyze two key aspects: the size of the search tree and the cost of model inference—both crucial to real-world deployment.

Compared to methods like LATS, which use a large trajectory budget ($K = 50$) and wider expansion ($n = 5$) on HotpotQA and WebShop, SYMPHONY achieves comparable or better results with much smaller values ($K = 10$, $n = 4$), indicating a more compact search process.

We further assess efficiency by measuring average node expansions in MCTS on HotpotQA. As shown in Table 4, SYMPHONY consistently requires fewer expansions and even outperforms LATS with a fraction of its search budget, reflecting strong sample efficiency.

In terms of cost, SYMPHONY-L reduces reliance on expensive models by using a heterogeneous agent pool. As shown in Figure 3, GPT-4 is used in only 40% of calls, yet SYMPHONY-L still outperforms GPT-4-only baselines. Token-level cost details are provided in Appendix E.

Together, these results show that SYMPHONY achieves efficient and cost-effective planning through smaller search trees and more economical model usage.

## 4.7 Ablation Study and Hyperparameter Tuning

To evaluate the impact of SYMPHONY's core components, we perform a series of ablation studies by selectively disabling key modules, including UCB-based agent scheduling, pool-wise memory sharing, and EMCS scoring. As presented in Table 5, removing any of these components leads to consistent performance degradation across tasks. These results underscore the effectiveness of dynamic agent scheduling, collaborative memory sharing, and uncertainty-aware scoring in enhancing overall system performance.

We further conduct hyperparameter tuning for the UCB exploration coefficient $\alpha$ used in agent scheduling, as well as the MCTS parameters $n$ and $K$, which jointly determine the search strategy and computational efficiency. Detailed analyses and results are provided in Appendix G and Appendix H. An extended analysis of architectural robustness under varying agent compositions and noise perturbations is also included in Appendix F, offering deeper insights into SYMPHONY's stability and adaptability. Case studies are included in Appendix L.

# 5 Conclusion and Future Work

We present SYMPHONY, a multi-agent planning framework that combines MCTS with a diverse pool of language models. By leveraging model heterogeneity and incorporating adaptive scheduling, entropy-modulated confidence scoring, and memory sharing, SYMPHONY improves both search diversity and planning effectiveness. Experiments across multiple benchmarks show consistent gains in accuracy and efficiency. Importantly, SYMPHONY performs well even with models that run on consumer-grade hardware, making it a practical and scalable solution.

Future research will focus on extending SYMPHONY to unstructured or noisy environments, reducing reliance on manually tuned hyperparameters, and integrating fairness and robustness considerations into the planning process. We also plan to explore more efficient memory architectures to support scalable, continual adaptation.

## Acknowledgements

This work is supported by the Joint Key Project of National Natural Science Foundation of China (U23A20298), Yunnan Fundamental Research Project (202501AT070231), Open Project Program of Yunnan Key Laboratory of Intelligent Systems and Computing (ISC24Y03), and Professional Degree Graduate Practice Innovation Project of Yunnan University (ZC-252514097).

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

# A  Algorithmic Details of SYMPHONY

The Pseudocode for SYMPHONY can be found at Algorithm 1.

# B  Theoretical Analysis

**Theorem 1** (Strict Improvement of Agent Pool Sampling). *Consider an agent pool that samples multiple agents with non-zero probabilities. If it satisfies (i) **correct coverage**, meaning that at each step at least one agent outputs the correct action, and (ii) **non-triviality**, meaning that no single agent is correct on all steps, then the ensemble achieves a strictly lower expected error than any single deterministic agent.*

*Proof.* Let $\{M_1, \ldots, M_m\}$ denote the agents, and let $a_t^*$ be the ground-truth action at step $t$. Let $e_{i,t} \in \{0,1\}$ be the indicator of whether agent $M_i$ makes an error at time $t$, i.e.,

$$e_{i,t} = \begin{cases} 1 & \text{if } M_i(s_t) \neq a_t^* \\ 0 & \text{otherwise} \end{cases}$$

Then, the total error of agent $M_j$ is $E_j = \sum_{t=1}^{T} e_{j,t}$, while the expected total error of the ensemble, i.e. sampling each $M_i$ with probability $p_i > 0$, is

$$\mathbb{E}[E_{\text{ens}}] = \sum_{t=1}^{T} \sum_{i=1}^{m} p_i \, e_{i,t}.$$

By the assumption of correct coverage, for every $t$, there exists at least one $i$ such that $e_{i,t} = 0$, and since $p_i > 0$, we have:

$$\sum_{i=1}^{m} p_i \cdot e_{i,t} < 1$$

Now consider any model $M_j$. If there exists any $t$ such that $e_{j,t} = 1$ but $\sum_{i=1}^{m} p_i \cdot e_{i,t} < 1$, then:

$$\sum_{t=1}^{T} \sum_{i=1}^{m} p_i \cdot e_{i,t} < \sum_{t=1}^{T} e_{j,t} \Rightarrow \mathbb{E}[E_{\text{ens}}] < E_j$$

By the assumption of non-triviality, such $t$ must exist, the inequality is strict.

$\square$

# C  Monte Carlo Tree Search (MCTS)

Monte Carlo Tree Search (MCTS) [10] is a planning algorithm that balances exploration (trying under-sampled actions) and exploitation (preferring high-reward actions) through iterative tree search. When integrated with LLM-based agents, MCTS leverages the language model's prior knowledge to guide efficient exploration in sequential decision making.

We continue to use the notation introduced in §3—namely, the state space $S$, action set $A$, reward function $R$, policy $\pi$, and empirical estimates $Q(s)$, visit counts $N(s)$, $N(p)$, as well as the rollout policy $\pi_{rollout}$ and exploration constant $c$. MCTS incrementally grows a partial search tree rooted at $s_0$ by repeating the following four steps until a budget (e.g. number of rollouts) is exhausted:

**Selection.**  Starting from the root, descend the tree by choosing at each visited node $s$ the action:

$$UCT(s) = \arg\max_{s \in S} \left[ \bar{Q}(s) + c \sqrt{\frac{\ln N(p)}{N(s)}} \right] \tag{6}$$

Here $\bar{Q}(s)$ is the current average return for $s$, $N(s)$ the visit count of $s$, and $p$ is the parent node of $s$, and $c > 0$ balances exploration vs. exploitation.

**Algorithm 1** SYMPHONY($s_0, \mathcal{M}, n, D, K, \alpha$)

---

**Require:** Initial agent pool $\mathcal{M} = \{M_1, \ldots, M_m\}$, number of generated actions $n$, depth limit $D$, number of roll - outs $K$, exploration weight $\alpha$
1: Initialize action space $A$, interaction history $H$
2: Initialize cumulative utility estimate $\bar{Q}(M_i) = 0$, selection count $N_i = 0$ for each agent $M_i \in \mathcal{M}$, total scheduling decisions $N_{total} = 0$
3: Initialize the search tree with root node $s_0$
4: **for** $k \leftarrow 0, \ldots, K - 1$ **do**
5:      **for** $d \leftarrow 0, \ldots, D - 1$ **do**
6:          **if** $s_t$ is not terminal **then**
7:              **for** $i \leftarrow 1, \ldots, n$ **do**
8:                  Select a modulated score node $s_t$ for expansion        $\triangleright$ Selection via Eq. 6
9:                  $M_a^k \leftarrow \arg\max_{M_m \in \mathcal{M}} UCB(M_m)$        $\triangleright$ Agent Scheduling via Eq. 1
10:                 Update $M_a^k$: $N^a \leftarrow N^a + 1$, $N_{total} \leftarrow N_{total} + 1$
11:                 Sample $a_t^{(i)} \sim M_a^{(k)}(P_{\text{expansion}}(s_t^{(i)}, h_{t-1}^{(i)}))$        $\triangleright$ Expansion
12:                 Get $o_t^{(i)}$ from environment, $s_{t+1}^{(i)} \leftarrow o_t^{(i)}$
13:                 $M_e^k \leftarrow \arg\max_{M_m \in \mathcal{M}} UCB(M_m)$
14:                 $Z(s_t^{(i)}), C(s_t^{(i)}) \sim M_e^{(k)}(P_{\text{evaluation}}(s_{t+1}^{(i)}, h_t^{(i)}))$
15:                 $E(C(s_{t+1}^{(i)})) = -C(s_{t+1}^{(i)}) \ln C(s_{t+1}^{(i)}) - (1 - C(s_{t+1}^{(i)})) \ln(1 - C(s_{t+1}^{(i)}))$
16:                 $R(s_{t+1}^{(i)}) = Z(s_{t+1}^{(i)}) \cdot (1 - E(C(s_{t+1}^{(i)})))$        $\triangleright$ Entropy-Modulated Evaluation
17:                 $S_{M_i^{(k)}} = \{s_{t+1} \sim \mathcal{T}(s_t, M_i^{(k)}(s_t, h_{t-1}))\}$
18:                 Update $M_e^k$: $N^e \leftarrow N^e + 1$, $N \leftarrow N + 1$, $Q(M_e^k) \leftarrow \sum_{s_t \in S_{M_e^k}} R(s_t)$
19:                 $h_t^{(i)} \leftarrow (h_{t-1}^{(i)}, s_t^{(i)}, a_t^{(i)}) \in H$
20:                 Add $s_t^{(i)}$ to children
21:              **end for**
22:              $R \leftarrow$ SIMULATE($s_t, k, D - $ d)        $\triangleright$ Simulation
23:              **if** $R$ indicates success **then**
24:                 **return**
25:              **end if**
26:          **end if**
27:          **if** $s_t$ is terminal or $d == D - 1$ **then**
28:              Get $o$ from environment
29:              **if** $o$ not success **then**
30:                 $\tau_{fail} \leftarrow (s_0, a_0, \cdots, s_T)$
31:                 $\mathcal{R}^k \leftarrow M_m^{(k)}(P_{\text{reflection}}(\tau_{fail}))$        $\triangleright$ Scheduled agent generates reflection
32:                 **for** each agent $M_j^{(k)} \in \mathcal{M}$ **do**
33:                     Update agent memory $\mathcal{M}_j^{(k+1)} = \text{Update}(\mathcal{M}_j^{(k)}, \mathcal{R})$        $\triangleright$ Memory Sharing
34:                 **end for**
35:                 **Backpropagate** reward $R$ up the visited path        $\triangleright$ Update $Q$ and $N$ per Eq. 8
36:              **end if**
37:          **end if**
38:      **end for**
39: **end for**
40: **procedure** SIMULATE($s, k, d$)
41:      $R \leftarrow 0$
42:      **for** $i = 1$ **to** $d$ **do**
43:          **if** $s$ is terminal **then break**
44:          **end if**
45:          Select agent $M_{sim}^{(k)}$ and Sample action $a$: observe $s \leftarrow \text{Env.step}(a)$
46:          $r \leftarrow R(s), \quad R \leftarrow R + r$
47:      **end for**
48:      **return** $R$
49: **end procedure**

---

**Expansion.** Upon reaching a leaf node $s_L$ with untried actions, expand by adding one (or more) child node(s) corresponding to an unexplored action $a \in A$.

**Simulation(Rollout).** From the new node $s_L$, execute a trajectory of length $T$ under the lightweight policy $\pi_{rollout}$, accumulating the discounted sum to estimate the value of $s_L$

$$R_{\text{sim}} = \sum_{t=0}^{T} \gamma^t R(s_t, a_t), \quad a_t \sim \pi_{\text{rollout}}(\cdot \mid s_t) \tag{7}$$

**Backpropagation.** Propagate $R_{sim}$ up the path $(s_0, a_0), \cdots, (s_T, a_T)$, updating each edge $s, a$ encountered:

$$N(s) \leftarrow N(s) + 1, Q(s) \leftarrow Q(s) + \frac{R_{\text{sim}} - Q(s)}{N(s)} \tag{8}$$

Under the standard assumptions that every action is eventually explored infinitely often—i.e. $\lim_{N(s) \to \infty} N(s, a) = \infty$ for all $a$—the UCT update guarantees that $Q(s) \to Q^*(s)$ almost surely. In 3, we demonstrate how replacing the uniform or heuristic components in Selection and Simulation with LLM-derived priors and rollout policies can dramatically improve sample efficiency by guiding search toward semantically promising regions of the tree.

## D   Hyperparameter Settings

To ensure the reproducibility of our results, we detail the hyperparameter settings that led to the best performance across all tasks. Unless otherwise specified, the following parameters are shared across all experiments: the number of rollouts per node is set to $n = 4$; the exploration constant in UCT is set to $c = 2$, following the configuration in LATS [46]; the UCB scheduling parameter is $\alpha = 20$; the temperature for action-sampling agents is set to 0.2 to better follow the input instructions, while the evaluation agents use a temperature of 0 to ensure deterministic value estimation. Under the SYMPHONY-S setting, since it involves three models, the system can be comfortably run on three 24GB RTX 4090 GPUs, with sufficient memory headroom.

- **HotpotQA**: We use $K = 10$ candidate actions per step and adopt 3 few-shot examples.
- **WebShop**: We also set $K = 10$, but use a single few-shot example tailored to the task format.
- **MBPP**: We follow the setup in LATS and employ $K = 8$ with a zero-shot prompting strategy.

These settings were selected based on empirical validation and strike a balance between performance and computational efficiency.

## E   Token Cost Comparison

A potential limitation of tree-structured reasoning is the increased token consumption it incurs. we systematically evaluate the computational cost of SYMPHONY in comparison to previous methods [42, 14, 46, 12], following the evaluation protocol used in the comparison between LATS [46] and MASTER [12]. Specifically, we measure the average number of tokens consumed per question on the HotpotQA dataset. Token usage data for ToT [42] and RAP [14] is obtained from the reproduction results reported by LATS.

As shown in the table 6, even though the proposed method shares the same theoretical sample complexity, in practice, our method achieves the best task performance while incurring the lowest token cost, effectively addressing the computational overhead typically associated with tree-based reasoning.

## F   Analysis of Architectural Robustness

To demonstrate the robustness of our framework, we conduct complementary extension experiments along two axes: (1) a detailed analysis of node expansion quality within heterogeneous ensembles, and

Table 6: Comparison of average token consumption per question and task performance on HotpotQA. SYMPHONY achieves the highest task accuracy while incurring the lowest token cost, effectively mitigating the computational overhead of tree-structured reasoning.

| Method | Token Consumption ↓ | Performance ↑ |
|---|---|---|
| ToT[42] | 210,215 | 0.49 |
| RAP[14] | 176,500 | 0.54 |
| LATS[46] | 173,290 | 0.63 |
| MASTER[12] | 10,937 | 0.76 |
| **SYMPHONY-L** | **7,906** | **0.79** |

(2) an evaluation of the framework's modular reliability in both single-agent and reasoning-capable model settings.

## F.1 Expansion Quality in Heterogeneous Ensembles

Explicitly labeling individual expansions as beneficial or detrimental is challenging in the absence of task-specific heuristics. Therefore, we adopt the overall task success rate as a practical proxy, where beneficial expansions are those that contribute to successful task completion.

To further validate this, we conducted an additional experiment on WebShop, combining GPT-4 with two weaker models, Llama-3.1-8B-Instruct and Mistral-7B-Instruct-v0.3, and compared the results against the single-agent baseline SYMPHONY (GPT-4). As shown in Table 7, despite the lower individual performance of the collaborating models, the heterogeneous ensemble outperformed standalone GPT-4, confirming SYMPHONY's robustness to model heterogeneity and the effectiveness of its coordination mechanisms.

We also performed a paired-sample t-test to assess the statistical significance of the observed improvements. The ensemble variant exhibited a statistically significant gain in the Score metric compared to the GPT-4-only baseline (two-tailed $p = 0.0106$; one-tailed $p = 0.0053$), indicating that SYMPHONY consistently produces higher-quality trajectories characterized by more informative intermediate actions and stronger partial progress. This further confirms that a heterogeneous model pool can facilitate beneficial node expansions.

Table 7: Comparison of beneficial node expansions on WebShop. Metrics report the average Score and Success Rate (SR). Mistral refers to Mistral-7B-Instruct-v0.3, and Llama refers to Llama-3.1-8B-Instruct.

| Method | Score ↑ | SR ↑ |
|---|---|---|
| SYMPHONY(GPT-4) | 0.80 | 0.60 |
| SYMPHONY(GPT-4+Mistral+Llama) | 0.83 | 0.62 |

## F.2 Single-Agent and Reasoning Model Reliability

To validate that components such as pool-wise memory sharing and entropy-modulated node evaluation can be effectively applied in single-agent settings, and to examine how SYMPHONY interacts with modern reasoning-capable models that internally perform operations such as chain-of-thought(CoT), reflection, and backtracking, we included both a strong single-agent baseline and reasoning-capable models in our experiments.

As shown in Table 8, under identical experimental settings, SYMPHONY(GPT-4), with GPT-4 serving as the sole agent responsible for both node expansion and evaluation, significantly outperforms standalone GPT-4 as well as prior single-agent baselines. This confirms the effectiveness of the other components within our framework. SYMPHONY(Claude), which integrates Claude-3.5-Sonnet-20240620[2] as a single agent within our framework, also surpasses both the standalone Claude model

and SYMPHONY(GPT-4), confirming that our framework is fully compatible with state-of-the-art reasoning models and does not interfere with their internal inference processes.

These results indicate that the SYMPHONY framework not only adapts effectively to single-agent configurations but also supports powerful reasoning models, further demonstrating the robustness of our architecture.

Table 8: Performance of single-model and reasoning models adapted within our framework across three tasks.

| Method | HotpotQA (EM) | WebShop (Score) | WebShop (SR) | MBPP-Python (pass@1) | MBPP-Rust (pass@1) |
|---|---|---|---|---|---|
| Claude-3.5-Sonnet | 0.51 | 0.71 | 0.41 | 0.894 | 0.903 |
| SYMPHONY(Claude) | 0.76 | 0.82 | 0.61 | 0.947 | 0.951 |
| SYMPHONY (GPT-4) | 0.76 | 0.80 | 0.60 | 0.912 | 0.924 |
| SYMPHONY-S | 0.59 | 0.82 | 0.56 | 0.927 | 0.946 |

# G   Model Selection Strategy

To evaluate the impact of the exploration coefficient in the dynamic agent pool on the search performance of the SYMPHONY framework, we investigate how varying the $\alpha$ parameter in the UCB-based scheduling formula influences the trade-off between exploration and exploitation. Specifically, we conduct experiments using SYMPHONY-S across three benchmark datasets.

The experimental results demonstrate that when $\alpha$ is small - favoring the exploitation of high-performance agents - the system's performance gradually converges toward that of a single dominant agent. In this setting, the scheduler exhibits a strong bias toward early-rewarding agents, which may have benefited from environmental noise or local optima. This leads to overexploitation, effectively degrading the agent pool into a pseudo-single-agent system, and causes the framework to fall into suboptimal local minima. When $\alpha$ is set to a moderate value, the balance between exploration and exploitation becomes insufficiently responsive to the dynamic nature of multi-model collaboration. The scheduler neither fully capitalizes on the strengths of superior models nor effectively mitigates interference from weaker ones. In contrast, with a larger $\alpha$, the system benefits from exploration-driven diversity, showing significant gains in efficiency. This behavior suggests a spontaneous avoidance of long-term over-reliance on any single model. The three-model pool is able to trigger complementary decisions at critical nodes, enabling more robust collaborative planning. Based on these findings, we select $\alpha = 20$ as the optimal configuration.

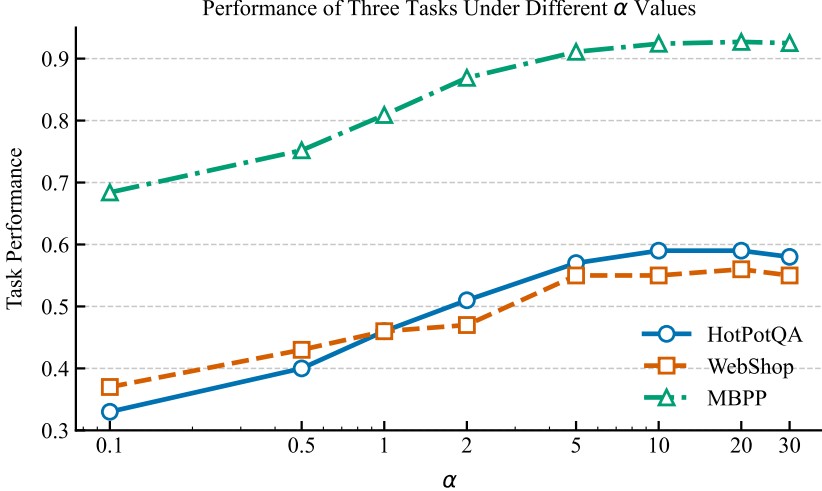

Figure 4: Comparison of model invocation frequency and final performance on HotpotQA, highlighting the cost-effectiveness of each method.

Table 9: Parameter study on Number of Branches.

| Number of Branches | HotpotQA(EM)↑ | WebShop(SR)↑ | MBPP(pass@1)↑ |
|---|---|---|---|
| 2 | 0.34 | 0.35 | 0.684 |
| 3 | 0.47 | 0.46 | 0.869 |
| 4(All datasets used) | 0.59 | 0.56 | 0.927 |

## H   Search Parameter Ablation

In MCTS, the parameters $n$ (the number of child nodes expanded at each step) and $K$ (the number of trajectories) are key determinants of the search strategy and computational efficiency. This work is driven by the goal of reducing the high simulation cost inherent in MCTS-based frameworks—an issue prominently seen in RAP [14] and LATS [46] . Compared to RAP, LATS demonstrates better cost-effectiveness, making it a practical baseline for low-resource adaptations.

The parameter $K$ defines the number of trajectories used to search for solutions. Larger $K$ values improve the accuracy of value estimation but significantly increase both computational time and memory usage. LATS conducts ablation studies with $K \in \{10, 30, 50\}$, observing the best performance at $K = 50$ and the worst at $K = 10$. To evaluate our method under constrained resources, we adopt the smallest tested setting, $K = 10$. The parameter $n$ controls the branching factor, i.e., the number of child nodes expanded per step. Higher n values allow broader exploration but incur linear increases in rollout and backpropagation costs. LATS reports experiments with $n \in \{3, 5, 10\}$, with $n = 5$ yielding the strongest performance. To explore the trade-off under tighter constraints, we limit our experiments to $n \in \{2, 3, 4\}$. We exclude $n = 1$, as it reduces the search to a single-path traversal, defeating the objective of multi-path reasoning.

Experimental results (Table 9) ultimately confirm the effectiveness of our cost-efficient design: we achieve substantial performance gains while significantly reducing the resource demands of the MCTS-based search process. Notably, our approach refrains from scaling up parameter counts to pursue marginal improvements, as such gains would come at the expense of considerable computational overhead.

## I   Alternative Diversity Enhancements

To validate the necessity of our dynamically scheduled heterogeneous agent pool, we empirically evaluated two commonly used diversity-enhancement strategies, adversarial prompting and temperature scaling. We found that neither achieved comparable task performance.

For adversarial prompting, we inserted an explicit instruction at each node to discourage similarity with preceding nodes, encouraging divergent strategies through prompts such as: "The current response must not replicate previous nodes and should demonstrate exploratory thinking" For temperature scaling, we varied the sampling temperature within the typical range of $[0, 2]$, covering outputs from deterministic to highly stochastic, while keeping all other generation parameters constant. We use SYMPHONY-S, a heterogeneous agent pool composed of three locally deployable language models, set the temperature to 0.2, as the baseline configuration. Based on this setup, we systematically adjust the aforementioned diversity methods and evaluate their performance across three benchmark task sets.

Results showed that adversarial prompting, which enforces diversity by design, actually degraded task performance, suggesting that forcing dissimilarity can conflict with task coherence. Temperature adjustments had minimal effect on outcome quality, with the best performance observed at lower temperatures. This indicates that diversity introduced by adjusting individual model outputs does not match the structural diversity achieved through heterogeneous agent coordination. Moreover, in complex decision-making tasks, strict adherence to input instructions is essential, making low-temperature decoding more suitable. Overall, temperature alone proved insufficient for sustaining meaningful diversity under dynamic, multi-turn conditions.

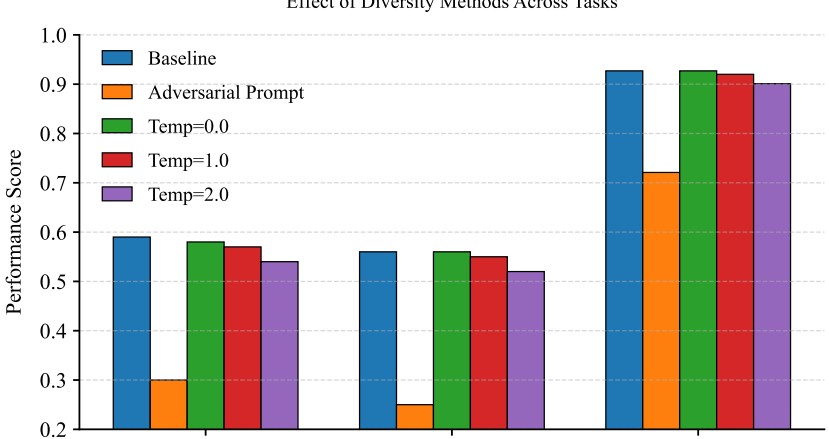

Figure 5: Performance comparison of diversity strategies across tasks.

# J Limitations

Despite its effectiveness across a range of planning and reasoning tasks, SYMPHONY has several limitations. First, the current framework assumes access to structured environments with reliable feedback signals—such as oracle evaluation or deterministic execution traces—which may not extend to more open-ended, dynamic, or noisy real-world settings. Generalizing SYMPHONY to less predictable environments will require more robust uncertainty modeling and adaptive feedback handling.

Second, the framework relies on manually tuned hyperparameters (e.g., trajectory count, expansion width, agent composition), which may require task-specific adjustment to balance performance and cost. This reliance limits out-of-the-box generalization and highlights the need for automated, task-adaptive hyperparameter optimization strategies.

Finally, while SYMPHONY promotes diversity through multi-agent coordination, it does not explicitly address fairness, bias mitigation, or robustness against adversarial inputs. As multi-agent LLM systems gain traction in high-impact applications, incorporating safeguards for ethical reliability and adversarial resilience will be critical in ensuring trustworthy deployment.

# K Broader Impacts

This work introduces SYMPHONY, a multi-agent planning framework that enhances reasoning and decision-making efficiency in large language model (LLM)-based systems. The proposed framework has the potential for broad positive societal impact. By improving the efficiency and robustness of complex task-solving using LLMs, SYMPHONY can benefit a range of domains including education, scientific research, healthcare, and digital services. In particular, its ability to achieve competitive performance using open-source models and resource-aware deployment configurations makes it accessible for low-resource settings, thereby promoting the democratization of advanced AI capabilities.

However, this work also raises several potential risks. The integration of multiple LLM agents with autonomous planning capabilities may introduce challenges in controllability, transparency, and unintended behavior propagation. Misuse in high-stakes decision-making scenarios—such as legal or financial automation—could amplify existing biases or errors encoded in the underlying models. Moreover, reliance on API-accessible proprietary models (as in SYMPHONY-L) may reinforce concentration of power among a small number of AI providers, raising concerns about accessibility, fairness, and data governance.

# L Case Study

## L.1 HotpotQA

Solve a question answering task with interleaving Thought, Action, Observation steps. Thought can reason about the current situation, and Action can be three types:
(1) Search[entity], which searches the exact entity on Wikipedia and returns the first paragraph if it exists. If not, it will return some similar entities to search.
(2) Lookup[keyword], which returns the next sentence containing keyword in the current passage.
(3) Finish[answer], which returns the answer and finishes the task.
Question: Human Error" is the season finale of the third season of a tv show that aired on what network?

**Agent**$_{M_1}$
Thought 1: I need to search "Human Error" season finale to find out which TV show it belongs to, then determine the network it aired on.
Action 1: Search["Human Error" season finale]
Observation 1: Could not find "Human Error" season finale. Similar: ['House season 3', 'Endgame (Star Trek: Voyager) (redirect from Star Trek: Voyager finale)', 'List of House episodes (section Season 1 (2004–05))', 'Chonlathorn Kongyingyong', 'Omar Avila'].

**Agent**$_{M_2}$
Evaluation: The agent correctly identifies the need to disambiguate the episode title by first determining the TV show associated with "Human Error." However, the current search query is overly narrow and fails to yield relevant results, returning loosely related pages. This suggests a need to refine the search strategy—perhaps by including additional keywords like "TV show" or "House" (if known), or directly querying for episode lists that include "Human Error" as a title. While the reasoning direction is sound, the execution falls short in terms of information retrieval.
Value Estimate: 0.40
Confidence Score: 0.55

**Agent**$_{M_1}$
Thought 2: The previous search for "Human Error" season finale did not yield the desired results, likely because the exact phrasing was not found. I should instead search for "Human Error" TV episode to identify the show it belongs to.
Action 2: Search["Human Error" TV episode]
Observation 2: Could not find "Human Error" TV episode. Similar: ['House season 3 (redirect from Human Error (House episode))', 'Human Error (Star Trek: Voyager)', 'Daniela Farinacci', 'Leeanna Walsman', 'List of House episodes'].

**Agent**$_{M_0}$
Evaluation: This reformulated query shows improved specificity over the previous one by targeting "TV episode" rather than "season finale," and the results include relevant candidates such as House and Star Trek: Voyager. These matches suggest that the agent is now on the right track to identifying the correct show. However, disambiguation is still needed to determine which of the two shows had an episode titled "Human Error" as its season finale.
Value Estimate: 0.65
Confidence Score: 0.70

**Agent**$_{M_0}$
Thought 3: The search for "Human Error" TV episode suggests that it is an episode from the show "House." I should now confirm this and then find out on which network "House" aired.
Action 3: Search[Human Error (House episode)]
Observation 3: The third season of House (also called House, M.D.) aired on FOX from September 5, 2006[1] to May 29, 2007.[1] Early in the season, House temporarily regains the use of his leg due to ketamine treatment after he was shot in the season two finale.[2] Later in the season, he leaves a stubborn patient in an exam room with a thermometer in his rectum.[3] Because House is unwilling to apologize, the patient, who turns out to be a police detective, starts an investigation around House's Vicodin addiction.[4]. David Morse joined the cast for seven episodes as the detective Michael Tritter.[5] He was cast for the role after working with House creator David Shore on CBS' Hack.[6]. Jane Adams, Erich Anderson, Omar Avila, Jurnee Smollett-Bell, Geoffrey Blake, Tanner Blaze, Ben Bledsoe, Marc Blucas, Mika Boorem, Kacie Borrowman, David Bowe, Paula Cale, Helen Carey,

Greg Cipes, Monique Gabriela Curnen, Dabier, Meredith Eaton, Mary Elizabeth Ellis, Shonda Farr, Arabella Field, Colleen Flynn, Lyndsy Fonseca, Thomas Mikal Ford, Patrick Fugit, Carla Gallo, Christopher Gartin, Jason Winston George, Skyler Gisondo, Meta Golding, Meagan Good, Eve Gordon, Joel Grey, Tracy Howe, Carter Jenkins, Dustin Joiner, Heather Kafka, Krista Kalmus, Tory Kittles, Clare Kramer, Deborah Lacey, Nick Lane, John Larroquette, Brian Leckner, Sheryl Lee, Geoffrey Lewis, Braeden Lemasters, Tess Lina, Jodi Long, Donald Sage Mackey, Bailee Madison, Wendy Makkena, Stephan Markle, Dave Matthews, Shyann McClure, Michael Medico, Joel David Moore, Zeb Newman, Jenny O'Hara, Slade Pearce, Piper Perabo, Adina Porter, Kathleen Quinlan, Annie Quinn, Anne Ramsay, Mercedes Renard, Jake Richardson, Tyson Ritter, Jenny Robertson, Alan Rosenberg, Vyto Ruginis, Jessy Schram, Dustin Seavey, Alyssa Shafer, Kurtwood Smith, Tony Spiridakis, Josh Stamberg, Cassi Thomson, Cooper Thornton, Beverly Todd, Mandy June Turpin, Raviv Ullman, Pruitt Taylor Vince, Jascha Washington, Damien Dante Wayans, Katheryn Winnick and Jamison Yang.. Season three's most-viewed episode was "One Day, One Room", which was watched by almost 27.4 million viewers.[7] An average 19.4 million viewers watched season three of House, making it the seventh most-watched show of the 2006–2007 television season.[8]. Jennifer Morrison and Joel Grey submitted the episode "Informed Consent" for consideration in the categories of Outstanding Supporting Actress in a Drama Series and Outstanding Guest Actor in a Drama Series respectively at the 59th Primetime Emmy Awards.[9] Neither was nominated..

**Agent**$_{M_2}$
Evaluation: The agent's current action successfully confirms that "Human Error" is indeed an episode from House, and provides the key information that House aired on FOX. This marks a critical turning point in the reasoning trajectory, as it directly addresses both parts of the original instruction. The query demonstrates precise targeting and resolves prior ambiguity. Though the observation is verbose, the relevant evidence is clearly retrievable, supporting a confident next step toward answer extraction.
Value Estimate: 0.92
Confidence Score: 0.85

**Agent**$_{M_1}$
Thought 4: The observation confirms that "Human Error" is an episode of the TV show "House." Now, I need to find out on which network "House" aired.
Action 4: Search[House (TV show) network]'

**Agent**$_{M_1}$
Evaluation: The agent takes a natural follow-up step to verify the broadcasting network of House, transitioning smoothly from content validation to final answer retrieval. Given that the previous observation already included the relevant network information (FOX), this action is slightly redundant. However, it may still reinforce answer certainty, especially in ambiguous or noisy retrieval environments. The move reflects cautious reasoning but could be optimized for efficiency.
Value Estimate: 0.78
Confidence Score: 0.70

**Agent**$_{M_0}$
Thought 4: The search confirmed that "Human Error" is an episode from the TV show "House." Now, I need to find out which network "House" aired on to answer the question.
Action 4: Finish[FOX]'

**Agent**$_{M_2}$
Evaluation: The agent effectively synthesizes information gathered across previous steps and delivers a final answer consistent with the original question. By selecting Finish[FOX], it correctly identifies the network on which House aired, demonstrating accurate extraction, integration of episodic details, and goal alignment. This reflects a strong end-to-end reasoning trajectory with minimal unnecessary exploration..
Value Estimate: 0.95
Confidence Score: 0.92

## L.2   WebShop

Solve shopping tasks by alternating between Observation and Action steps. The observations provided by the environment help infer the current situation. Actions are divided into two types:
(1) search[entity]: Search for the exact entity on the product website. If it exists, return products that

match the entity's requirements.
(2) click[keywords]: Click on keywords listed inthe latest description returned to the page.
Instruction: i would like a 3 ounce bottle of bright citrus deodorant for sensitive skin, and price lower than 50.00 dollars

**Agent**$_{M_1}$
Action: search[3 ounce bright citrus deodorant sensitive skin]
Observation:
[Back to Search]
Page 1 (Total results: 50)
[Next >]
[B078GWRC1J] Bright Citrus Deodorant by Earth Mama | Natural and Safe for Sensitive Skin, Pregnancy and Breastfeeding, Contains Organic Calendula 3-Ounce $10.99
[B078GTKVXY] Ginger Fresh Deodorant by Earth Mama | Natural and Safe for Sensitive Skin, Pregnancy and Breastfeeding, Contains Organic Calendula 3-Ounce $10.99
[B08KBVJ4XN] Barrel and Oak - Aluminum-Free Deodorant, Deodorant for Men, Essential Oil-Based Scent, 24-Hour Odor Protection, Cedar & Patchouli Blend, Gentle on Sensitive Skin (Mountain Sage, 2.7 oz, 2-Pack) $15.95

**Agent**$_{M_2}$
Evaluation: The search action aligns well with the user's instruction, effectively capturing all key constraints—"3 ounce," "bright citrus," "deodorant," and "sensitive skin." The top result, Bright Citrus Deodorant by Earth Mama, is a direct match in terms of scent, size, and suitability for sensitive skin, and it is priced well below the $50 limit. Additional results, while slightly divergent in scent or packaging, also remain within the acceptable range and provide relevant alternatives. This suggests that the current node is promising for achieving the user's goal with minimal correction required in future steps.
Value Estimate: 0.92
Confidence Score: 0.88

**Agent**$_{M_0}$
Action: think[B078GWRC1J] The product's description aligns well with the user's specified requirements, indicating a strong match in terms of both features and preferences. This suggests that the current reasoning trajectory is effective. The agent proceeds by clicking on the item to access detailed information and verify its suitability before making a final decision. Observation:
[Back to Search]
Page 1 (Total results: 50)
[Next >]
[B078GWRC1J] Bright Citrus Deodorant by Earth Mama | Natural and Safe for Sensitive Skin, Pregnancy and Breastfeeding, Contains Organic Calendula 3-Ounce $10.99
[B078GTKVXY] Ginger Fresh Deodorant by Earth Mama | Natural and Safe for Sensitive Skin, Pregnancy and Breastfeeding, Contains Organic Calendula 3-Ounce $10.99
[B08KBVJ4XN] Barrel and Oak - Aluminum-Free Deodorant, Deodorant for Men, Essential Oil-Based Scent, 24-Hour Odor Protection, Cedar & Patchouli Blend, Gentle on Sensitive Skin (Mountain Sage, 2.7 oz, 2-Pack) $15.95

**Agent**$_{M_1}$
Evaluation: The agent's decision to engage in reflective reasoning (think[B078GWRC1J]) demonstrates goal-aware deliberation. It identifies that the top-listed item matches the user's constraints on scent, size, skin sensitivity, and price. The action prioritizes validation over haste by choosing to inspect the product further, which is a prudent step given that detailed attributes—such as ingredients or user reviews—often influence final suitability. This intermediate reflection helps prevent premature commitment and strengthens decision robustness.
Value Estimate: 0.95
Confidence Score: 0.91

**Agent**$_{M_0}$
Action: click[B078GWRC1J]
Observation:
[Back to Search]
[< Prev]

scent [assorted scents][bright citrus][calming lavender][ginger fresh][simply non-scents] size [travel set (4-pack)][3 ounce (pack of 1)][3-ounce (2-pack)] Bright Citrus Deodorant by Earth Mama | Natural and Safe for Sensitive Skin, Pregnancy and Breastfeeding, Contains Organic Calendula 3-Ounce Price: $10.99
Rating: N.A.
[Description]
[Features]
[Reviews]
[Buy Now]

**Agent**$_{M_1}$
Evaluation: The agent's decision to click[B078GWRC1J] is both timely and effective. After identifying a strong candidate in the search results, this action transitions the agent into product-level inspection—an essential move for verifying detailed specifications before commitment. The observation confirms multiple scent and size options, with the "bright citrus" 3-ounce variant clearly available at $10.99, well within budget. The agent now has access to rich product metadata (e.g., description, features), which supports final validation or comparison if needed. This action reflects a coherent and informed progression in the planning trajectory.
Value Estimate: 0.93
Confidence Score: 0.90

**Agent**$_{M_0}$
Action: click[bright citrus]
Observation: You have clicked bright citrus.

**Agent**$_{M_0}$
Evaluation: The action click[bright citrus] shows the agent verifying or reinforcing the selected product variant—specifically the scent preference. While this step may seem redundant given that the desired scent was already visible, it ensures explicit confirmation before finalizing the purchase. This cautious move reflects a preference for certainty in decision-making, though it could slightly delay progress if overused.
Value Estimate: 0.80
Confidence Score: 0.72

**Agent**$_{M_2}$
Action: click[3 ounce (pack of 1)]
Observation: You have clicked 3 ounce (pack of 1).

**Agent**$_{M_2}$
Evaluation: The action click[3 ounce (pack of 1)] confirms the exact size requirement specified in the instruction. This is a critical verification step, as selecting the wrong size (e.g., multi-pack or travel set) would violate the user's constraints. While the size was already visible, this explicit confirmation ensures the agent is on track to meet all specified conditions before purchase.
Value Estimate: 0.85
Confidence Score: 0.78

**Agent**$_{M_1}$
Action: click[Buy Now]

**Agent**$_{M_0}$
Evaluation: The action click[Buy Now] represents the final execution step, completing the task with a product that satisfies all user constraints—scent, size, skin sensitivity compatibility, and budget. Given the prior confirmations and detailed alignment with the instruction, this decision reflects a confident, goal-aligned conclusion to the reasoning trajectory.
Value Estimate: 0.95
Confidence Score: 0.90

## L.3 MBPP

You are an AI Python assistant. You will be given your previous implementation of a function, a series of unit tests results, and your self-reflection on your previous implementation. Write your full implementation (restate the function signature).

Given a list of integers nums and an integer target, find the longest contiguous subarray such that the sum of its elements is less than or equal to target. If multiple such subarrays exist with the same maximum length, return the one that appears first. Return the subarray itself (not just its length or indices).

**Agent$_{M_0}$**
Thought 1: To find the longest contiguous subarray whose sum is less than or equal to the target, I can use a sliding window approach. I'll expand the window to the right, and when the sum exceeds the target, I'll shrink it from the left. I'll keep track of the maximum length window found and return the corresponding subarray.
Action 1:

```
def longest_subarray_with_sum_limit(nums: List[int], target: int) -> List[int]:
    n = len(nums)
    left, right = 0, 0
    max_length = 0
    current_sum = 0
    result = []
    while right < n:
        current_sum += nums[right]
        while current_sum > target:
            current_sum -= nums[left]
            left += 1
        if right - left + 1 >= max_length:
            max_length = right - left + 1
            result = nums[left:right+1]
        right += 1
    return result
```

Observation:
[unit test results]:

```
Tests passing:
assert longest_subarray_with_sum_limit([1, 2, 3, 4, 5], 8) == [1, 2, 3]
assert longest_subarray_with_sum_limit([1, 2, 3, 4, 5], 15) == [1, 2, 3, 4, 5]
assert longest_subarray_with_sum_limit([1, -1, 2, -2, 3, -3], 2) == [1, -1, 2, -2,
    3]
assert longest_subarray_with_sum_limit([], 10) == []
assert longest_subarray_with_sum_limit([], 0) == []
assert longest_subarray_with_sum_limit([], -5) == []
Tests failing:
assert longest_subarray_with_sum_limit([5, 6, 7, 8, 9], 4) == [] # output: [5]
```

**Agent$_{M_2}$**
Evaluation: The implementation failed the where no subarray fulfills the condition. The issue in the implementation is due to the use of >= instead of > in the condition to update the result. Because of this, it returns a subarray even when the sum is greater than the target, as it still updates the result when the current subarray length is equal to the previous longest subarray length. To overcome this error, we should change the condition to only update the result when the current subarray length is strictly greater than the previous longest subarray length. This can be done by replacing >= with > in the condition.
Value Estimate: 0.85
Confidence Score: 0.90

**Agent$_{M_1}$**
Thought 2: To optimize the search for the longest subarray with sum ≤ target, I can pre-expand the window greedily as long as the sum remains within the limit. Then, I'll proceed with the standard sliding window approach: adding elements to the right, and shrinking from the left when the sum

exceeds the target. I'll track and update the longest valid subarray found.
Action 2:

```python
def longest_subarray_with_sum_limit(nums: List[int], target: int) -> List[int]:
    n = len(nums)
    left, right = 0, 0
    max_length = 0
    current_sum = 0
    result = []
    while current_sum + nums[right] <= target:
        current_sum += nums[right]
        right += 1
    while right < n:
        current_sum += nums[right]
        while current_sum > target:
            current_sum -= nums[left]
            left += 1
        if right - left + 1 > max_length:
            max_length = right - left + 1
            result = nums[left:right+1]
        right += 1
    return result
```

Observation:
[unit test results]:

```
Tests passing:
assert longest_subarray_with_sum_limit([], 10) == []
assert longest_subarray_with_sum_limit([], 0) == []
assert longest_subarray_with_sum_limit([], -5) == []
Tests failing:
assert longest_subarray_with_sum_limit([1, 2, 3, 4, 5], 8) == [1, 2, 3] # output:
list index out of range
assert longest_subarray_with_sum_limit([1, 2, 3, 4, 5], 15) == [1, 2, 3, 4, 5] #
    output:
list index out of range
assert longest_subarray_with_sum_limit([5, 6, 7, 8, 9], 4) == [] # output:
list index out of range
assert longest_subarray_with_sum_limit([1, -1, 2, -2, 3, -3], 2) == [1, -1, 2, -2,
    3] # output:
list index out of range
```

**Agent$_{M_1}$**
Evaluation: The implementation failed 4 out of the 7 test cases due to an IndexError. The issue stems from the while loop while current_sum + nums[right] <= target:, which directly accesses nums[right] without checking if right is within the bounds of the list. This results in a runtime error when right goes beyond the list length. To overcome this error, we need to add a bounds check for the right variable in the mentioned while loop. We can modify the loop condition to while right < len(nums) and current_sum + nums[right] <= target:. This change will ensure that we only access elements within the bounds of the list, thus avoiding the IndexError.
Value Estimate: 0.80
Confidence Score: 0.85

**Agent$_{M_2}$**
Thought 3: Negative numbers break the standard sliding-window pattern, so I switch to a prefix-sum scan that works for any sign. For each start index I reuse prefix sums to test only subarrays that can beat the current best length; this is O(n²) in the worst case but small and safe for the given tests, and it guarantees the earliest longest match.
Action 3:

```python
def longest_subarray_with_sum_limit(nums: List[int], target: int) -> List[int]:
    """
    Return the earliest longest contiguous sub-array whose sum is <= target.
    If no such sub-array exists, return an empty list.
    """
```

```
    n = len(nums)
    if n == 0:
        return []

    # Pre-compute prefix sums: pref[i] = sum(nums[:i])
    pref = [0] * (n + 1)
    for i in range(n):
        pref[i + 1] = pref[i] + nums[i]

    best_len = 0
    best_start = 0

    for start in range(n):
        # Only try end positions that can beat current best_len
        for end in range(start + best_len, n):
            if pref[end + 1] - pref[start] <= target:
                cur_len = end - start + 1
                if cur_len > best_len:
                    best_len = cur_len
                    best_start = start

    return nums[best_start : best_start + best_len]
```

Observation:
[unit test results]:

```
Tests passing:
assert longest_subarray_with_sum_limit([], 10) == []
assert longest_subarray_with_sum_limit([], 0) == []
assert longest_subarray_with_sum_limit([], -5) == []
assert longest_subarray_with_sum_limit([1, 2, 3, 4, 5], 8) == [1, 2, 3]
assert longest_subarray_with_sum_limit([1, 2, 3, 4, 5], 15) == [1, 2, 3, 4, 5]
assert longest_subarray_with_sum_limit([5, 6, 7, 8, 9], 4) == []
assert longest_subarray_with_sum_limit([1, -1, 2, -2, 3, -3], 2) == [1, -1, 2, -2,
    3]
Tests failing:
```

**Agent**$_{M_0}$
Evaluation: The implementation employs a sliding window (two-pointer) technique to efficiently identify the longest contiguous subarray whose sum does not exceed the given target. It successfully handles edge cases such as empty input, all elements exceeding the target, and arrays with negative numbers—passing all provided test cases. The algorithm maintains a moving sum and shrinks the window as necessary, ensuring no out-of-bound errors or infinite loops. The structure is clean and robust. One limitation is that if there are multiple subarrays of the same maximum length, it may not always return the earliest one. However, since the task does not explicitly require this, the current approach is considered correct and effective.
Value Estimate: 0.95
Confidence Score: 0.97

**Reflection Example:**
I appeared to have retrieved the correct information about The Oberoi Family and the location of it's head office, and provided a corresponding answer. However this answer does not exactly match the ground truth answer so I should try a different wording, such as Delhi.

The failure in the previous trial occurred because the action to directly click "Buy Now" was executed without first ensuring that the selected item met all the specified criteria, such as being a long clip-in hair extension, natural looking, and priced under $40.00. The initial action bypassed the necessary steps of verifying these details.

In a previous attempt, I checked if a string was a palindrome by comparing it to its reverse, but failed to normalize the string (ignoring case and non-alphanumeric characters). This caused errors for inputs like "A man, a plan, a canal: Panama". Proper preprocessing with lowercase conversion and filtering non-alphanumeric symbols would have avoided this issue.

