# OpenReview forum: "SYMPHONY: Synergistic Multi-agent Planning with Heterogeneous Language Model Assembly"
_NeurIPS.cc/2025/Conference — NeurIPS 2025 poster_

### Official Review · Reviewer_QwDw · 2025-06-18

**Clarity:** 3
**Significance:** 3
**Originality:** 3
**Rating:** 4
**Confidence:** 4

**Summary:**

SYMPHONY introduces a synergistic multi-agent planning framework that integrates heterogeneous language models to enhance Monte Carlo Tree Search (MCTS) for complex planning tasks. The framework addresses single-agent limitations by employing multiple diverse language models that collaborate across all four MCTS phases: selection, expansion, simulation, and backpropagation. It offers flexible deployment configurations (SYMPHONY-S for consumer hardware, SYMPHONY-L for cloud APIs) with a modular architecture including agent manager, coordination engine, MCTS planner, and evaluation modules. Experimental results demonstrate significant performance improvements over traditional single-agent methods, establishing a new paradigm for LLM-based autonomous planning through collective intelligence.

**Questions:**

See  Weaknesses.

**Ethical Concerns:**

["NO or VERY MINOR ethics concerns only"]

**Final Justification:**

The authors have addressed my concerns in their rebuttal, and I will maintain my positive score.

**Limitations:**

yes

**Quality:**

3

**Strengths And Weaknesses:**

Strengths

1. Novel Problem Formulation：
The paper addresses a novel and significant problem by identifying the limitations of single-agent MCTS approaches in complex planning tasks. The introduction of heterogeneous multi-agent collaboration represents a fresh perspective that hasn't been systematically explored in LLM-based planning before.

2. Clear and Comprehensible Presentation：
The paper is well-written with clear explanations of complex concepts, making the methodology and technical details accessible to readers. The logical flow from problem identification to solution design and experimental validation enhances overall readability and understanding.

3. Strong Alignment Between Motivation and Experimental Analysis：
The experimental results strongly support the paper's core motivation, particularly demonstrated in Figure 2 where the diversity differences between methods are significant and clearly visible. This empirical evidence directly validates the theoretical premise that multi-agent approaches provide superior exploration diversity.

4. Efficient Method with Practical Deployment Considerations：
The proposed method demonstrates superior efficiency compared to baseline approaches while offering practical value through dual configurations. SYMPHONY-S (consumer-grade hardware) and SYMPHONY-L (cloud API) provide flexible deployment options that align with real-world application scenarios and resource constraints.


Weaknesses

1. Limited Real-World Agent Scenario Alignment：
The experimental datasets have a significant gap compared to real agent scenarios, with environments like WebArena being more aligned with practical use cases. However, given that the authors follow the strongly related baseline work MASTER (which also adopts the same datasets), this is more of a personal concern rather than a major flaw in the experimental design.

2. Insufficient Analysis of Node Expansion Quality：
The experiments lack deeper analysis regarding the quality of diverse node expansions - specifically, whether beneficial versus detrimental expansions are strongly correlated with model capabilities. This becomes particularly important when there are significant capability gaps between different models in the heterogeneous ensemble.

3. Missing Single Strong Model Baseline：
An additional baseline should be supplemented since components like Pool-wise Memory Sharing and Entropy-Modulated Node Evaluation are actually applicable to single agents. The authors should include experimental results of a single powerful model within their proposed framework for fair comparison.

4. Unclear Effectiveness with Reasoning Models：
It remains uncertain whether the experimental conclusions hold for reasoning models, as these models already perform MCTS-like operations during long chain-of-thought processes, including reflection and backtracking. This raises questions about whether such capabilities might diminish the value of the proposed multi-agent approach.

---

> ### Author Rebuttal · Authors · 2025-07-31
>
> Thank you for your thoughtful and constructive feedback. We respond point-by-point to each of your concerns and questions below, and we deeply appreciate the opportunity to clarify and expand on the key aspects of our work.
>
>
> ### Weakness 1: Limited Real-World Agent Scenario Alignment
> Thank you for raising this valuable concern. We fully agree that aligning evaluation tasks with real-world agent scenarios is crucial for practical impact, and we genuinely appreciate your suggestion of incorporating environments like WebArena. This is an excellent direction, and we’re glad you brought it to our attention.
>
> We recognize that WebArena, with its open-ended and dynamic web environment, offers a more realistic approximation of real-world agent usage. Its task structure involving multimodal interactions, tool usage, and exploration provides a compelling platform for testing generalist agents in complex scenarios.
>
> That said, our current experimental suite consists of HotpotQA, WebShop, MBPP-Python, and MBPP-Rust, and was carefully selected to cover diverse and challenging reasoning demands across multiple domains.
>
> ● HotPotQA: Multi-hop question answering requiring fact chaining and intermediate reasoning.
>
> ● WebShop: Simulates goal-driven dialogue and sparse reward navigation over structured catalog environments.
>
> ● MBPP (Python & Rust): Program synthesis under strict correctness constraints, reflecting structured output reasoning and functional validation.
>
> While these tasks may lack the visual or interactive complexity of WebArena, they nonetheless capture essential characteristics of real-world agent behavior, such as long-horizon planning, reasoning under uncertainty, and memory across steps.
>
> We wholeheartedly agree that extending SYMPHONY to more grounded and interactive environments is an important next step. Based on your suggestion, we plan to include WebArena or similar environments in future iterations of our framework, and we believe our method’s modularity and memory design make it a natural fit.
>
> We sincerely appreciate this insightful feedback. It encourages us to further extend the scope of SYMPHONY, and we look forward to pursuing this direction in future work.
>
> ### Weakness 2: Insufficient Analysis of Node Expansion Quality
>
> Thank you for raising this important point regarding the quality of node expansions in relation to model capabilities within a heterogeneous ensemble.
> We address this concern from two perspectives:
>
> First, although explicitly labeling expansions as “beneficial” or “detrimental” is challenging without task-specific heuristics, we use overall task success rate as a practical proxy. Beneficial expansions are those leading toward successful task completion. Our results show a clear correlation between model capability and expansion quality—larger, more capable models (e.g., GPT-4) tend to produce more effective expansions, reflected in higher success rates.
>
> Second, SYMPHONY’s design inherently mitigates risks from capability imbalances. The system prevents weaker models from dominating by combining:
>
> ● UCT-driven branch selection that favors high-value paths,
>
> ● UCB-based agent routing that downweights agents with historically poorer performance,
>
> ● Tree-structured search with backtracking that limits repeated exploitation of suboptimal nodes.
>
> This architecture ensures robustness against noisy or poor expansions. While “one bad apple” may exist, the tree search allows strong agents to reinforce better branches and dilute negative influence.
>
> To further support this, we conducted an additional experiment on WebShop combining GPT-4 with two weaker models (Llama-3.1-8B-Instruct and Mistral-7B-Instruct-v0.3). Despite the weaker collaborators, the ensemble outperformed standalone GPT-4, confirming SYMPHONY’s robustness to heterogeneity and the effectiveness of our coordination mechanisms.
>
> > | WebShop               | Score | SR   |
> > | :-------------------- | :---- | :--- |
> > | SYMPHONY(GPT-4) | 0.80  | 0.60 |
> > | SYMPHONY(GPT-4+Mistral+Llama)   | 0.83  | 0.62 |
>
>
> ### Weakness 3: Missing Single-Strong-Model Baseline
>
> Thank you for this valuable suggestion. We agree that components like Pool-wise Memory Sharing and Entropy-Modulated Node Evaluation can be applied in single-agent settings, and we have accordingly added a strong single-agent baseline for fair comparison.
>
> Specifically, we incorporate GPT-4, a widely adopted and powerful general-purpose model, as the sole agent within the full SYMPHONY framework across all three datasets. This ensures a meaningful and equitable baseline since GPT-4 is also used by many recent methods.
>
> ● The experiments run the identical framework as the multi-agent setup, but with GPT-4 as the only agent responsible for node expansion and evaluation.
>
> ● Even as a single model, our framework significantly improves over raw GPT-4 and prior single-agent baselines, demonstrating the value of our pool-wise memory and entropy-modulated evaluation mechanisms.
>
> ● Nonetheless, as discussed in our response to Weakness 2, the heterogeneous multi-agent SYMPHONY configuration still surpasses this enhanced single-agent baseline, confirming that model diversity yields complementary strengths unattainable by any individual model alone.
>
> > |                       | HotpotQA(EM) | WebShop(Score) | WebShop(SR) | MBPP-Python(pass\@1) | MBPP-Rust(pass\@1) |
> > | :-------------------- | :----------- | :------------- | :---------- | :------------------- | :----------------- |
> > | LATS | 0.71         | 0.76           | 0.38        | 0.811                | -             |
> > |MASTER | 0.76         | 0.80           |-      | 0.910                | -             |
> > | SYMPHONY(GPT-4) | 0.76         | 0.80           | 0.60        | 0.912                | 0.924              |
> > | SYMPHONY-L | 0.79         | 0.88           | 0.72        | 0.965               | 0.974              |
>
> These findings underscore the modularity and effectiveness of our approach in both single-agent and multi-agent contexts. We will include these new results in the revision to address your suggestion.
>
>
> ### Weakness 4: Unclear Effectiveness with Reasoning Models
>
> Thank you for this thoughtful and insightful question. We appreciate your concern regarding how SYMPHONY interacts with modern reasoning-capable models that internally perform operations like chain-of-thought (CoT), reflection, and backtracking.
> Our detailed response is as follows:
>
> ● Focus on Cost-Efficient Test-Time Scaling: The primary motivation behind SYMPHONY is to achieve competitive performance comparable to large, cloud-based models—but at significantly lower cost—by coordinating a diverse ensemble of lightweight agents at test time without heavy pretraining or fine-tuning. This approach emphasizes scalability and practicality, showing that structured multi-agent coordination and reflection mechanisms enable smaller models to collectively reach strong  performance.
>
> ● Experimental Fairness and Controlled Comparison: While reasoning-tuned models naturally exhibit stronger task performance, most baselines we compare against (e.g., RAP, LATS, MASTER) do not incorporate such specialized models. To maintain fair and controlled comparisons, we intentionally excluded these powerful reasoning models from our main experiments, preventing inflated baselines that could obscure SYMPHONY’s architectural contributions.
>
> ● Verifying Compatibility with Reasoning-Capable Models: To address your concern, we conducted an additional experiment integrating Claude-3.5-Sonnet-20240620, a reasoning-enhanced agent, as a single agent within our framework
>
> > |                       | HotpotQA(EM) | WebShop(Score) | WebShop(SR) | MBPP-Python(pass\@1) | MBPP-Rust(pass\@1) |
> > | :-------------------- | :----------- | :------------- | :---------- | :------------------- | :----------------- |
> > | Claude-3.5-Sonnet | 0.51         | 0.71           | 0.41        | 0.894                |0.903              |
> > | SYMPHONY(Claude-3.5-Sonnet) | 0.76         | 0.82           | 0.61        | 0.947                | 0.951              |
> > | SYMPHONY(GPT-4) | 0.76         | 0.80           | 0.60        | 0.912                | 0.924              |
> > | SYMPHONY-S | 0.59         | 0.82           | 0.56        | 0.927                | 0.946              |
>
>
> SYMPHONY (Claude) outperformed both the standalone Claude model and SYMPHONY (GPT-4) under identical settings, confirming that our framework is fully compatible with state-of-the-art reasoning models and does not interfere with their internal inference capabilities. Moreover, SYMPHONY-S, built on lightweight agents, also surpassed Claude-3.5-Sonnet, aligning with our core motivation of achieving strong task performance through efficient test-time coordination of smaller models.
>
> More importantly, this demonstrates that SYMPHONY’s benefits extend beyond ensembles of weaker agents, offering potential for hybrid setups that coordinate multiple specialized reasoning agents to further improve decision quality.
>
> We sincerely thank the reviewer for this valuable suggestion, which helped us showcase the generality and extensibility of our approach beyond the original scope.
>
> We sincerely appreciate your careful review. We hope our clarifications have resolved your concerns, and we would be happy to engage in further discussion if needed.

---

> > ### Comment · Reviewer_QwDw · 2025-08-07
> >
> > Thank you for your rebuttal and the new experimental results. Regarding weakness #2, the performance difference between SYMPHONY (GPT-4) and SYMPHONY (GPT-4 + Mistral + Llama) appears relatively small. Have you conducted any statistical significance tests to confirm whether this improvement is consistent and meaningful?
> >
> > Additionally, could you provide some concrete, illustrative cases study where the heterogeneous ensemble (GPT-4 + Mistral + Llama) demonstrably outperforms the single-agent GPT-4 setup—particularly in terms of node expansion, exploration diversity, or recovery from early errors? This would help clarify how the multi-agent collaboration contributes to better overall performance despite the smaller models' individual limitations.

---

> > > ### Author Response · Authors · 2025-08-07
> > > **Reply 1**
> > >
> > > Thank you for your thoughtful follow-up. Due to the response length constraint, we have split our reply into two consecutive comments. Please note that the content is logically continuous across both.
> > >
> > > ### Statistical Significance Analysis
> > > Thank you for raising the question of statistical significance. While the improvement in Success Rate (SR) between SYMPHONY (GPT-4) and our full ensemble (GPT-4 + Mistral + Llama) is not statistically significant, the difference in Score is statistically robust and consistent across tasks.
> > >
> > > We conducted paired-sample t-tests on the WebShop benchmark to assess significance. The ensemble model shows a statistically significant improvement in Score over the GPT-4-only baseline (two-tailed p = 0.0106; one-tailed p = 0.0053), indicating that it consistently generates higher-quality trajectories with more informative intermediate actions and better partial progress.
> > >
> > > The lack of significance in SR is expected and does not reflect a weakness in our method. SR is a coarse, binary metric that overlooks meaningful progress unless the final step exactly matches the ground truth. In contrast, Score provides a fine-grained, path-sensitive evaluation aligned with human utility. For example, a partially correct trajectory that recovers from early mistakes or provides useful partial results contributes to Score but not SR.
> > >
> > > These results confirm that the performance gains introduced by our heterogeneous model pool are both consistent across tasks and statistically meaningful, validating the effectiveness of our design under rigorous evaluation.
> > >
> > > ### Case Study
> > >
> > > To further illustrate the strengths of our heterogeneous ensemble, we present a detailed case study on the WebShop task:
> > >
> > > User query: _“I am looking for a grey sectional sofa for my living room”_
> > >
> > > As shown in the table, under the GPT-4-only configuration with tree width n=4, each expansion layer produces near-identical actions, reflecting a collapse in output diversity. In decision-making tasks with strict instruction constraints, the model's inherent stochasticity is substantially suppressed, leading to identical input-output mappings across nodes and causing MCTS to degenerate into a single linear search path. In this case, after an initial failure when exploring “grey” products, the model shifts its attention to other dimensions such as price or rating, but remains unsuccessful. This failure cascade results in a cognitive deadlock: the model repeatedly issues the click[Back to Search] action without forming new hypotheses, ultimately failing to escape the loop.
> > >
> > > In the heterogeneous ensemble configuration, MCTS regains its intended role as a structured and breadth-aware search process. Initially, the system encounters a similar cognitive deadlock as in the single-agent GPT-4 setting. However, with the introduction of diverse candidate actions, components like Pool-wise Memory Sharing and Entropy-Modulated Node Evaluation begin to take effect. These mechanisms guide the search away from previously failed trajectories and toward alternative yet semantically relevant products, including variations in attributes such as color.
> > >
> > >
> > > While GPT-4 remains the dominant decision-maker through Agent Scheduling, the integration of smaller models injects critical behavioral diversity into the search process. These alternative proposals, though not always optimal on their own, provide enough exploratory momentum to help the system break out of local failure modes. The final success still stems from GPT-4, but only after the ensemble has expanded the search space enough to surface a more accurate hypothesis.

---

> > > > ### Author Response · Authors · 2025-08-07
> > > > **Reply 2**
> > > >
> > > > > | Step           | SYMPHONY(GPT-4)                                                                             | SYMPHONY(GPT-4+Mistral+Llama)                               |
> > > > > |:---------------|:--------------------------------------------------------------------------------------------|:------------------------------------------------------------|
> > > > > | **Query**      | _search[grey sectional sofa]_ * 4                                                           | **G**: _search[grey sectional sofa]_ * 2 (Selected)         |
> > > > > |                |                                                                                             | **M**: _search[grey modular sofa for living room]_          |
> > > > > |                |                                                                                             | **L**: _search[modern grey sectional sofa for living room]_ |
> > > > > | **Filter**     | _click[B09GVDTN7S]_ * 4                                                                     | **G**: _click[B09GVDTN7S]_ * 3 (Selected)                   |
> > > > > |                |                                                                                             | **M**: _click[B01NADEYMV]_                                  |
> > > > > | **Check**      | _click[dark grey]_ * 4                                                                      | **G**: _click[dark grey]_ * 2 (Selected)                    |
> > > > > |                |                                                                                             | **M**:  _click[light grey]_                                  |
> > > > > |                |                                                                                             | **L**: _click[Buy Now]_                                     |
> > > > > | **Attempt**    | _click[Buy Now]_  * 4 (reward=0.67)                                                         | **G**:_click[Buy Now]_ * 3 (reward=0.67)                    |
> > > > > |                |                                                                                             | **L**: _click[Buy Now]_                                     |
> > > > > | **Reflection** | ...such as budget, size, or additional features, were not accounted for in the analysis.... | While the sofa matches the style and color requirements...  |
> > > > > | **Filter**     | _click[B01NADEYMV]_ * 4                                                                     | **G**: _click[B01NADEYMV]_ * 2 (Selected)                   |
> > > > > |                |                                                                                             | **M**: _click[Back to Search]_                              |
> > > > > |                |                                                                                             | **L**: _click[Next >]_                                      |
> > > > > | **Attempt**    | _click[Buy Now]_  * 4 (reward=0.67)                                                         | **G**:_click[Buy Now]_ * 4 (reward=0.67)                    |
> > > > > | **Final**      | ..._search[...]_ ...  _click[Back to Search]_...  **Error**                                 | ..._click[light grey]_... **Success**                       |
> > > >
> > > > In the table, G denotes content generated by GPT-4, M by Mistral-7B-Instruct-v0.3, and L by Llama-3.1-8B-Instruct. In this particular task, the user’s actual preference aligns only with the “light grey” version of the product, despite having initially specified “grey.” The earlier failures were not due to logical errors, but to premature fixation on dark grey variants that semantically matched the query yet failed to satisfy the user’s true intent.
> > > >
> > > > This case illustrates a broader strength of the ensemble approach: in tasks involving ambiguous or underspecified user preferences, heterogeneous model integration enables MCTS to maintain exploration breadth and recover from early missteps. When paired with supporting components, it allows the system to converge more quickly on successful outcomes that single-agent configurations consistently miss.
> > > >
> > > > This example also illustrates the original inspiration behind our design: the collapse of tree diversity under homogenized strategies, which degrades multi-path exploration into a single linear trajectory. As discussed in our response to other reviewers, we experimented with adversarial prompting and temperature scaling to artificially induce diversity, but these methods resulted in unstable, incoherent behaviors and reduced task performance. In contrast, our multi-agent setup elicits natural, semantically grounded divergences, differences that stem from model-specific inductive priors rather than stochastic perturbations. This allows MCTS to reclaim its structural advantages, enabling disambiguation and robust decision making without sacrificing coherence.
> > > >
> > > > Thank you again for your valuable insights. If you have further questions or comments, we welcome continued discussion and look forward to your response.

---

> > > > > ### Comment · Reviewer_QwDw · 2025-08-08
> > > > >
> > > > > Thank you to the authors for your response, which has fully addressed my concerns, especially with the help of concrete case studies. I have one additional question: from the current results, it appears that the role of weaker models is primarily to introduce diversity into the system—an alternative to techniques like adversarial prompting and temperature scaling. Now, if we were to replace all models in the framework with strong, similarly capable ones—such as Gemini, GPT, and Claude—would this setting better highlight the advantages of your method? And what new or interesting experimental phenomena do you think might emerge in such a scenario?

---

> > > > > > ### Author Response · Authors · 2025-08-08
> > > > > >
> > > > > > Thank you for your follow-up. Your question aligns closely with a direction we were eager to explore during our research. As described in our paper, we designed two configurations: SYMPHONY-S (targeted at consumer-grade hardware) and SYMPHONY-L (leveraging large-scale foundation models via cloud-based APIs). SYMPHONY-L consists of a heterogeneous model pool of GPT-4, Qwen-Max (2024-09-19), and DeepSeek-V3 (2025-03-24). According to their public reports, Qwen-Max and DeepSeek-V3 have achieved performance on certain benchmarks that is comparable to GPT-4.
> > > > > >
> > > > > > In the WebShop environment, we compared SYMPHONY-L, SYMPHONY-S, SYMPHONY (GPT-4+Mistral-7B+Llama-8B), and SYMPHONY (GPT-4). We observed that SYMPHONY-S, even with smaller models, achieved performance already close to that of stronger models. When replacing all models in the SYMPHONY framework with high-capability and similarly performing models, SYMPHONY-L achieved a significant performance boost, making the advantages of our method even more pronounced. This advantage is likewise reflected in the main results across all datasets presented in the paper.
> > > > > >
> > > > > > > |                                      | WebShop(Score) | WebShop(SR) |
> > > > > > > |:-------------------------------------|:---------------|:------------|
> > > > > > > | SYMPHONY (GPT-4)                     | 0.80           | 0.60        |
> > > > > > > | SYMPHONY (GPT-4+Mistral-7B+Llama-8B) | 0.83           | 0.62        |
> > > > > > > | SYMPHONY-S                           | 0.82           | 0.56        |
> > > > > > > | SYMPHONY-L                           | 0.88           | 0.72        |
> > > > > >
> > > > > > As shown in Figure 3 of our paper, there are substantial differences in API costs per 1M generated tokens: GPT-4 costs 60.00, Qwen-Max costs 1.32, and DeepSeek-V3 costs 1.10. For other baselines such as ToT, RAP, LATS, and MASTER, which rely solely on GPT-4, the costs are extremely high. In our method, since the three models have no substantial performance gap, our Agent Scheduling mechanism can dynamically allocate decisions across them, using lower-cost models to offset the expenses of higher-cost models. This cost-effectiveness is also reflected in Table 4 of the paper and Table 6 in the appendix, where our method achieves the best results in task performance, token consumption, and memory usage.
> > > > > >
> > > > > > In summary, a stronger and capability-balanced model pool can further highlight the advantages of our approach, enabling more powerful and efficient MCTS search while outperforming other baselines at a lower cost.
> > > > > >
> > > > > > We are glad to have this exchange, as your question captures our original motivation and experimental thinking. We welcome any further discussion you may wish to have.

---

### Official Review · Reviewer_aqqm · 2025-07-02

**Clarity:** 3
**Significance:** 3
**Originality:** 3
**Rating:** 4
**Confidence:** 3

**Summary:**

This paper proposes SYMPHONY, a multi agent framework with different models that utilize a variation of Monte Carlo Tree Searching planning.  The authors adopt heterogeneous models to prevent narrow rollouts. The authors evaluate SYMPHONY on HotPotQA, WebShop and MBPP, showing promising results on all three benchmarks.

**Questions:**

Q1: Line 1236 and afterwards, "Refelcion" -> "Reflection"?

Q2: How big is the Reflection buffer for each dataset and each experiment of SYMPHONY? Are all agents sharing the same reflections?

Q3: Is there any better way in agent selection instead of averaging?

**Ethical Concerns:**

["NO or VERY MINOR ethics concerns only"]

**Final Justification:**

During rebuttal, the authors explained why choosing the Bernoulli Distribution under Entropy-Modulated Confidence Scoring (EMCS). I value this explanation more than others.

The authors explained the settings of reflections, and pointed out future directions in agent selection.

Thus, I maintain my rating (4).

**Limitations:**

Yes

**Quality:**

3

**Strengths And Weaknesses:**

Strength:
1. Using multiple models to diversify rollouts and decision trajectories is simple yet effective.

2. The manipulation on $Q$ term of UCT is clever, mainly based on Entropy-Modulated node evaluation, which is simple yet effective, making the scoring dynamic based on each agent as a scorer for itself. Such scores are reflected dynamically on difficulty of various tasks. The average of multiple agents to reflect on $Q$ term is straightforward.

3.The section of efficiency and cost analysis is essential to assess such multi agent framework, and SYMPHONY shows great efficiency in terms of token consumption.

4. Experiments set up are detailed and fairly complete.

Weakness:
1. It would be great if more explanation is provided about why choosing the Bernoulli Distribution under Entropy-Modulated Confidence Scoring (EMCS).

---

> ### Author Rebuttal · Authors · 2025-07-31
>
> ### Weakness 1: It would be great if more explanation is provided about why choosing the Bernoulli Distribution under Entropy-Modulated Confidence Scoring (EMCS).
>
> Thank you for highlighting this important aspect. Our choice to use the Bernoulli distribution in Entropy-Modulated Confidence Scoring (EMCS) is driven by the need for a principled and efficient way to calibrate agent confidence across heterogeneous models.
>
> In SYMPHONY, each agent assigns a scalar confidence score $c \in (0, 1)$ to a candidate node’s value estimate. Since agents vary in architecture, training data, and scale, their raw confidence scores are not directly comparable or equally reliable.
>
> To address this, we interpret ccc as the success probability of a Bernoulli distribution and compute its Shannon entropy:
> $$
> H(c) = -c \ln c - (1 - c) \ln(1 - c)
> $$
>
> This entropy quantifies the epistemic uncertainty of the agent’s estimate:
>
> ● It peaks at $c = 0.5$, reflecting maximum uncertainty.
>
> ● It approaches zero as $c \to 0$  or $c \to 1$, indicating high confidence.
>
> We then modulate the original confidence score by this entropy, effectively downweighting uncertain (high-entropy) scores and emphasizing confident (low-entropy) ones. This modulation requires no additional hyperparameters and integrates seamlessly into node evaluation.
>
> The Bernoulli entropy is a natural choice because:
>
> ● It is a well-established, theoretically grounded measure of uncertainty in probabilistic decision-making.
>
> ● Its symmetry and strict convexity ensure consistent and smooth handling of both overconfident and underconfident scores.
>
> ● It is computationally efficient, avoiding complex resampling or ensemble methods, which is crucial for real-time MCTS.
>
> Empirically, as demonstrated in Table 5, EMCS improves node evaluation reliability amid noisy or divergent outputs from heterogeneous agents. It enables SYMPHONY to balance trust in decisive agents with caution toward uncertain ones, enhancing overall multi-agent search performance.
>
> ### Question 1: Line 1236 and afterwards, "Refelcion" -> "Reflection"?
>
> Thank you for catching this. The term in that section should indeed be “Evaluation” rather than “Reflection.” It refers to the process where an agent evaluates and assigns a score to a node, often providing a brief explanation or rationale.
>
> Since we do not employ a separate reasoning model for scoring, the agent performs a lightweight verbal evaluation using natural language. This approach enhances the stability and interpretability of the score by grounding it in reasoning, rather than relying on isolated scalar values. We will correct this terminology in the final version.
>
> ### Question 2: How big is the Reflection buffer for each dataset and each experiment of SYMPHONY? Are all agents sharing the same reflections?
>
> Thank you for this excellent question. We are happy to clarify the details regarding the reflection buffer in SYMPHONY.
>
> The reflection buffer is scoped at the task level, meaning it is associated with each individual MCTS search tree corresponding to a single task instance—for example, one shopping session in WebShop or a single question in HotpotQA. It is not shared or accumulated across the entire dataset.
>
> During the search process for each task, we maintain a fixed-size FIFO buffer containing the three most recent reflections. This size balances memory efficiency with sufficient representational capacity to capture meaningful failure insights. Once the search for that task concludes, whether successfully or not, the buffer is cleared because its role is to influence decision-making only within the context of that specific task’s trajectory.
>
> All agents share this reflection buffer. This design enables heterogeneous agents to benefit from collective verbal memory of prior failures, supporting our hypothesis that diverse models provide complementary perspectives and improve reasoning when informed by shared reflective feedback.
>
> Additionally, the inherent MCTS backpropagation mechanism reduces the likelihood of revisiting suboptimal trajectories, so only a modest reflection buffer is necessary. Reflections act as verbal reinforcement that complements rather than replaces MCTS’s intrinsic credit assignment.
>
> We also note that related work, such as Reflexion (Shinn et al., NeurIPS 2023) , demonstrates that increasing reflection frequency and buffer size can enhance performance up to a saturation point. Investigating the impact of buffer size and retention strategies on planning quality is an important avenue for future research.
>
> In summary, the reflection buffer is task-specific, modest in size, shared across agents, and plays a supportive role in improving multi-agent collaboration within each discrete task.
>
>
> ### Question 3: Is there any better way in agent selection instead of averaging?
>
> Thank you for this insightful and forward-looking suggestion.
>
> Currently, SYMPHONY employs a vanilla UCB strategy that uses mean reward estimates to balance exploration and exploitation among agents. This choice is driven by several factors:
>
> ● Simplicity and robustness: UCB is a well-established, parameter-free, and interpretable method that integrates naturally with the recursive structure of MCTS.
>
> ● Strong empirical support: Our ablation study (Table 5) shows that removing or replacing UCB, such as using round-robin or uniform sampling, leads to a notable drop in accuracy.
>
> That said, your suggestion opens promising avenues for future research. Incorporating weighting schemes based on recency or contextual similarity could enable more adaptive agent routing, particularly in environments with concept drift or non-stationarity.
>
> We are keen to investigate more sophisticated selection methods, including softmax-weighted UCB, Bayesian UCB, or contextual bandits, to further enhance flexibility and adaptivity in multi-agent coordination.
>
> Thank you again for your valuable input, which aligns well with SYMPHONY’s vision of dynamic and effective multi-agent collaboration.
>
> We sincerely appreciate your careful review. We hope our clarifications have resolved your concerns, and we would be happy to engage in further discussion if needed.
>
> ### Reference:
> Shinn, N., Cassano, F., Gopinath, A., Narasimhan, K., & Yao, S. (2023). Reflexion: Language agents with verbal reinforcement learning. Advances in Neural Information Processing Systems, 36, 8634–8652.

---

> > ### Comment · Reviewer_aqqm · 2025-08-01
> >
> > Thank you for addressing my questions. I will maintain my rating.

---

> > > ### Author Response · Authors · 2025-08-02
> > >
> > > Thank you for taking the time to review our paper. We truly appreciate your thoughtful comments. Your suggestions will be incorporated into the next revision, and we believe they will significantly strengthen the final version.
> > >
> > > Thank you again for your support.

---

### Official Review · Reviewer_AH1d · 2025-07-04

**Clarity:** 3
**Significance:** 3
**Originality:** 3
**Rating:** 3
**Confidence:** 2

**Summary:**

Existing single‐agent planning approaches lack of structural diversity constrains exploration, leading to redundant search trajectories and degraded planning quality. To address this, the paper introduce SYMPHONY (SYnergistic Multi‐agent Planning with HeterOgeneous laNguage model assemblY), a multi‐agent MCTS framework that orchestrates a diverse pool of language‐model agents. By assigning different models to propose and evaluate actions, SYMPHONY substantially increases rollout diversity and enables more robust exploration. Across three challenging benchmarks, SYMPHONY delivers outperforming existing state-of-the-art baselines.

**Questions:**

Do all agents serve the same function? If so, in what way are they “heterogeneous” rather than merely independent copies of the same model?

How are next states determined? Is each agent queried on a separate environment copy in parallel?

What exactly is stored in “pool-wise memory”? Natural-language reflections, embeddings, or something else? How do agents access and use this memory during search? How does memory differ from the set of past trajectories?

Missing Related Work: For example, “Monte Carlo Planning with Large Language Model for Text-Based Game Agents” (ICLR 2025)

**Ethical Concerns:**

["NO or VERY MINOR ethics concerns only"]

**Limitations:**

See weaknesses.

**Quality:**

3

**Strengths And Weaknesses:**

Strengths

The research problem is compelling: leveraging multiple agents to enrich planning diversity.

The paper evaluates SYMPHONY with several off-the-shelf LLM APIs for fair comparison.

Experiments use benchmarks such as HotpotQA, WebShop, and MBPP cover reasoning, sequential decision-making, and code generation.

Weaknesses

It’s unclear what is fundamentally new: beyond assembling multiple models, which components are original contributions?

Why is a straightforward UCB bandit strategy (balancing per-agent invocation counts and rewards) sufficient? What insights guided this choice?

How does SYMPHONY’s runtime and memory usage compare to single-agent MCTS or existing multi-agent methods?

---

> ### Author Rebuttal · Authors · 2025-07-31
>
> Thank you for your careful reading and thoughtful questions. We appreciate the opportunity to clarify several aspects of SYMPHONY and highlight its conceptual and empirical contributions.
>
> ### Weakness 1: what is fundamentally new?
>
> Thank you for your question. The fundamental novelty of SYMPHONY lies in its heterogeneous expert pool, which is a carefully curated collection of structurally and pretraining-diverse open-source LLMs such as Qwen, Mistral, and Llama. Unlike ensembles of homogeneous or similarly configured models, this diverse pool leverages architectural differences to produce complementary reasoning behaviors. Importantly, by coordinating multiple smaller open-source models deployable on consumer-grade hardware, SYMPHONY achieves performance that approaches or even surpasses that of the much larger, closed-source GPT-4.
>
> The other core mechanisms, namely the UCB-based expert scheduling, the shared runtime reflection memory, and the entropy-modulated node evaluation, are all designed specifically to support and enhance this heterogeneous model pool. The UCB scheduler adaptively routes search efforts among agents to balance exploration and exploitation efficiently. The reflection memory stores natural language summaries of failure cases to share cautionary insights across agents, guiding reasoning away from known pitfalls in real time. Meanwhile, the entropy-modulated evaluation selectively preserves useful candidate expansions while filtering out misleading ones without expensive rollouts.
>
> Together, these components form a tightly integrated decision-making framework built around the heterogeneous expert pool, enabling SYMPHONY to deliver robust, high-quality performance beyond what standard homogeneous ensembles or majority voting can achieve.
>
> ### Weakness 2: Why is a straightforward UCB bandit strategy (balancing per-agent invocation counts and rewards) sufficient? What insights guided this choice?
>
> Thank you for the question. We chose a UCB1-style bandit mechanism for agent selection based on several key insights.
>
> First, agent selection in SYMPHONY naturally maps to a multi-armed bandit problem, where balancing exploration and exploitation over time is crucial. UCB offers a principled, low-overhead solution with strong theoretical regret guarantees and asymptotic fairness among agents.
>
> Second, compared to learned routing methods that require additional training and complexity, UCB is stateless, parameter-free, and highly interpretable. This simplicity suits the recursive, high-frequency decision process in MCTS and does not interfere with convergence properties.
>
> Third, our empirical results (Table 5) demonstrate that removing UCB-based scheduling significantly degrades performance, confirming that this seemingly simple strategy is essential for effective coordination among heterogeneous agents.
>
> Finally, the reward signals，derived with EMCS, are often sparse and non-stationary due to partial observability. UCB naturally handles such variability, maintaining robustness in uncertain environments.
>
>
> ### Weakness 3: SYMPHONY’s runtime and memory usage are compared to single-agent MCTS and existing multi-agent methods
>
> Thank you for your question. We provide a detailed analysis of SYMPHONY’s runtime and memory usage in Section 4.6 and Appendix Table 6 to ensure a thorough and fair comparison.
>
> First, all methods were tuned with comparable tree parameters to enable equitable runtime evaluation. As shown in Table 4, SYMPHONY achieves higher task success rates while generating fewer tokens and expanding fewer search nodes than strong single-agent baselines such as ToT, RAP, and LATS.
>
> Second, we compare SYMPHONY against MASTER, a recent state-of-the-art multi-agent method, using their reported metrics since their code is unavailable. Appendix Table 6 shows SYMPHONY uses fewer nodes and fewer tokens, confirming superior runtime and memory efficiency.
>
> Because all methods are runtime frameworks based on tree-structured search, the cost can be measured transparently by the total number of expanded nodes and total model-generated tokens. From a runtime perspective, SYMPHONY uses smaller tree hyperparameters and terminates earlier under identical task budgets, leading to faster and more efficient search. From a memory perspective, the explicit count of visited nodes serves as a fair proxy.
>
> It is true that hosting SYMPHONY requires maintaining multiple models rather than a single one. However, by leveraging multiple open-source models deployable on consumer-grade hardware, SYMPHONY outperforms GPT-4-level performance without requiring prohibitively expensive infrastructure. Meanwhile, its more compact MCTS search tree further reduces computational overhead compared to single-agent approaches.
>
> In summary, SYMPHONY balances hosting cost with runtime and memory efficiency to deliver both more effective and more efficient task solving compared to single-agent or existing multi-agent methods.
>
> ### Question 1: Heterogeneity in agent pool
>
> Thank you for your question. While all agents share the same interface and perform the same high-level function within SYMPHONY, their fundamental differences lie in their reasoning styles, stemming from their architecture and pretraining methodology. We select models from different organizations, such as Qwen, Mistral, and Llama, that vary in network design, training data, tokenizer vocabularies, and alignment objectives.
>
> These intrinsic differences lead each agent to follow distinct reasoning paths and exhibit unique inductive biases. It is this heterogeneity in reasoning patterns, not merely having multiple copies of the same model, that introduces the meaningful diversity we rely on. Our framework leverages this reasoning-mode heterogeneity to explore complementary solution subspaces, improving robustness and overall task performance.
>
> ### Question 2: How are next states determined? Is each agent queried on a separate environment copy in parallel?
>
> Thank you for your question. SYMPHONY maintains a single environment instance, with explicit environment states saved at each node of the MCTS tree to support branching and backtracking.
>
> At each expansion step, a subset of agents is selected via the UCB scheduler. Each agent independently proposes a candidate action, generating multiple hypothetical child nodes. These candidates are then evaluated collectively using Entropy-Modulated Confidence Scoring (EMCS) combined with UCT, and only one child node is selected for further exploration.
>
> Since SYMPHONY operates in text-based environments, environment transitions are simulated by saving and restoring interaction histories (i.e., textual state). Each MCTS node stores its corresponding environment state snapshot, enabling efficient re-entry or rollback to any prior state without needing multiple environment copies running in parallel.
>
> This approach ensures accurate and memory-efficient environment simulation during search, while supporting sequential agent queries rather than parallel environment instances.
>
> ### Question 3: Memory Mechanism
>
> Thank you for your question. The pool-wise memory stores natural-language reflections, which are concise verbal summaries generated exclusively after failed MCTS rollouts. These reflections provide diagnostic feedback to guide and improve future decision-making.
>
> Each reflection is a high-level natural language description of a failure case. For example, in the WebShop task, a reflection might be: “Filtered by color before narrowing down by category — may miss relevant items.”
>
> The memory is implemented as a fixed-size FIFO buffer holding the three most recent reflections per MCTS episode (i.e., per task-specific search tree). This design ensures runtime efficiency and prevents the accumulation of stale or irrelevant reflections.
>
> When an agent is selected to propose candidate actions at a node, the current pool-wise memory—the latest reflections—is injected into the agent’s prompt. This provides contextual warnings or reminders that help the agent avoid repeating past mistakes.
>
> Importantly, this memory differs fundamentally from a set of past trajectories or embeddings. It does not store raw trajectory data nor embeddings, nor does it involve parameter updates. Instead, it acts as a lightweight, natural-language augmentation that enables on-the-fly adaptation during test time without additional training.
>
> This mechanism allows SYMPHONY to dynamically capture failure patterns and share them across heterogeneous agents during search, enhancing robustness while maintaining low computational overhead.
>
> ### Question 4: Missing Related Work
>
> Thank you for bringing this relevant work to our attention. The cited paper addresses single-agent memory-augmented MCTS for text-based games, sharing the goal of enhancing decision-making through memory mechanisms. However, there are key differences between their approach and SYMPHONY.
>
> Specifically, SYMPHONY proposes a multi-agent decision routing framework coordinating heterogeneous LLMs, while the cited work focuses on a single-agent setting. Moreover, our memory mechanism employs natural-language reflections dynamically injected at runtime to guide reasoning, in contrast to their stored trajectory-level memory used for retrospective lookback.
>
> We note that this paper was posted on arXiv on April 23, 2025, at 16:23 UTC, just days before the NeurIPS 2025 submission deadline. According to NeurIPS guidelines, works appearing on public servers within March 1st, 2025 prior to submission are considered concurrent. Due to the timing and our extensive experimental commitments, we were unable to reference it in the original submission. We appreciate this helpful pointer and will include a thorough discussion and citation in the final revision.
>
> We sincerely appreciate your careful review. We hope our clarifications have resolved your concerns, and we would be happy to engage in further discussion if needed.

---

> > ### Comment · Reviewer_AH1d · 2025-08-04
> >
> > Thanks for the response. In the presence of states and interaction histories, a standard UCB approach may not fully address the complexities of dynamic environments.

---

> > > ### Author Response · Authors · 2025-08-04
> > >
> > > Thank you for the follow-up. We fully agree that dynamic environments with evolving states and interaction histories pose substantial challenges beyond what standard bandit formulations assume. In fact, designing more sophisticated routing mechanisms that explicitly leverage state representations and long-term dependencies is an exciting direction we are actively exploring.
> > >
> > > That said, we were also initially surprised by how well UCB1 performed in our setting—significantly outperforming baselines across benchmarks. Upon deeper reflection, we believe this effectiveness stems from UCB's ability to strike a robust balance between exploration and exploitation without requiring explicit modeling of environment dynamics. This property proves particularly useful under sparse and non-stationary reward signals, which frequently arise due to partial observability and asynchronous interactions.
> > >
> > > Moreover, it's worth noting that UCB variants have long been successfully applied in structured decision-making under uncertainty. For instance, the Upper Confidence Bound applied to Trees (UCT) algorithm, central to Monte Carlo Tree Search (MCTS), is itself a direct extension of UCB, and was a core component of AlphaGo's planning strategy (Silver et al., 2016). Our use of UCB for agent routing follows a similar intuition: if UCB-based scheduling works well for node-level selection in deep search trees, then adapting it to the agent level (where each agent proposes node expansions) can serve as a lightweight yet surprisingly effective alternative to heavier learned controllers.
> > >
> > > We appreciate your suggestion and agree that a deeper understanding of UCB's behavior in relation to state-awareness deserves future investigation.
> > >
> > > ### Reference:
> > >
> > > Silver, D., Huang, A., Maddison, C. J., Guez, A., Sifre, L., Van Den Driessche, G., Schrittwieser, J., Antonoglou, I., Panneershelvam, V., Lanctot, M., et al. (2016). Mastering the game of Go with deep neural networks and tree search. Nature, 529(7587), 484–489.

---

### Official Review · Reviewer_y451 · 2025-07-15

**Clarity:** 3
**Significance:** 3
**Originality:** 3
**Rating:** 4
**Confidence:** 3

**Summary:**

This paper looks into a system called SYMPHONY that tries to improve planning and reasoning by using not just one, but a mix of different language models. The idea is that these models, since they're trained differently and might “think” in slightly different ways, can complement each other when solving complex tasks.

Instead of sticking to a single LLM, SYMPHONY brings together a pool of models and lets them work through problems collectively. There’s a mechanism that decides which model to use at each step based on how well it’s done in the past (a kind of UCB strategy). When a plan doesn’t work, the system writes out a short natural language “reflection” of what went wrong, and shares that with all the models. It also includes a scoring method that accounts for how confident the models seem about their outputs.

The authors tested this on three very different tasks: answering multi-hop questions (HotpotQA), making decisions in a simulated shopping scenario (WebShop), and writing code (MBPP). Across the board, SYMPHONY did better than well-known baselines like ToT and Reflexion—not just in terms of getting the right answers, but also using fewer tokens.

All in all, the paper makes a good case that combining several diverse models and letting them share feedback can improve both accuracy and efficiency, without needing extra training or tuning.

**Questions:**

Q1. In your experience, have there been any cases where bringing in more diverse models actually made things worse? I’m curious if there’s ever a point where the added heterogeneity introduces too much noise or conflict, instead of helping the system.

Q2. Can you share any real examples of what the reflections look like? Even just a snippet or two would help me understand how these reflections affect what the agents do next, and how meaningful or consistent they tend to be.

Q3. Roughly how often does the reflection mechanism come into play during rollouts? I’m trying to get a feel for whether it’s a frequent part of the process or something that only kicks in when an agent fails.

Q4. What size of memory buffer did you use for storing reflections? And did you happen to try out different sizes to see if that had any noticeable effect on performance?

Q5. Have you noticed any particular types of scenarios where SYMPHONY doesn’t work as well? For instance, in messier tasks where the signals are noisy or the goals aren’t well-defined, does it tend to struggle there?

**Ethical Concerns:**

["NO or VERY MINOR ethics concerns only"]

**Final Justification:**

I sincerely thank the authors for their detailed explanation. Most of my concerns have been addressed, and I will keep my original score.

**Limitations:**

Yes

**Quality:**

3

**Strengths And Weaknesses:**

Strengths

- SYMPHONY is tested on three very different tasks—HotpotQA (for multi-hop QA), WebShop (decision-making), and MBPP (writing code in Python and Rust). Across all of them, it does better than well-known methods like ToT, Reflexion, LATS, MASTER, and AgentCoder. The authors also run the tests with different random seeds, which helps make the results more trustworthy.

- The paper is easy to follow. Each part of SYMPHONY is explained with math, diagrams, and simple descriptions. The figures and tables help a lot, so even if you’re not an expert, it’s not too hard to understand how the system works.

- The way they use UCB to pick which model to call during search makes sense—it balances trying new models with sticking to the ones that work. Also, instead of retraining anything, agents just share short natural language reflections. That’s a simple but smart way to help them learn from each other.


Weaknesses

- The paper shows strong results, but it doesn’t talk much about the theory behind how SYMPHONY works—especially when using multiple agents and random rollouts. There’s no clear explanation of whether the system is stable or always gets better with more steps.

- Reflections are a big part of this method, but the paper doesn’t show any actual examples. It’s hard to know how useful they are without seeing what they look like or how consistent they are.

- There’s not much discussion about when SYMPHONY might fail. For example, what if agents give very different answers, or if one agent shares a reflection that misleads the others? These kinds of issues aren’t really addressed, which could be important in harder or noisy tasks.

---

> ### Author Rebuttal · Authors · 2025-07-31
>
> We sincerely thank the reviewer for these thoughtful and constructive comments. Below, we provide detailed responses to each of the raised concerns, covering theoretical soundness, reflection examples and usage, diversity trade-offs, and limitations in noisy tasks.
>
> ### Weakness 1: On Theoretical Understanding: Why Should SYMPHONY Work?
>
> To theoretically analyze the benefit of multi-agent coordination in sequential decision-making, we consider a setting in which multiple large language models (LLMs) are deployed as independent agents. At each decision step, one agent is sampled from a fixed distribution and is responsible for selecting an action. We assume that each agent may make mistakes on certain inputs, but that the agent pool collectively satisfies a correct coverage property: at every step, there exists at least one agent capable of choosing the correct action.
>
> Formally, Consider a sequential decision task with $T$ steps and a set of $m$ LLMs $\\{M_1, M_2, \\dots, M_m\\}$. At each step $t$, let $a_t^{\*}$ be the correct action. Assume that for every step $t$, there exists at least one model $M_i$ such that $M_i(s_t) = a_t^{\*}$. Then, for any probability distribution $\\{p_1, \\dots, p_m\\}$ with $p_i > 0$ and $\\sum_i p_i = 1$, the ensemble model that samples $M_i \\sim p_i$ at each step has strictly lower expected total error than any individual model in the ensemble:
>
> $$
> \\mathbb{E}[E_{\\mathrm{ens}}] < \\min_{j \\in [m]} E_j
> $$
> Let $e_{i,t} \\in \\{0,1\\}$ be the indicator of whether model $M_i$ makes an error at time $t$, i.e.,
> $$
> e_{i,t} =
> \begin{cases}
> 1 & \text{if } M_i(s_t) \ne a_t^* \\\\
> 0 & \text{otherwise}
> \end{cases}
> $$
> Then the total error of model $M_j$ is
> $$
> E_j = \sum_{t=1}^T e_{j,t}
> $$
> The ensemble model samples $M_i \sim p_i$ at each step, and its expected total error is
> $$
> \mathbb{E}[E_{\mathrm{ens}}] = \sum_{t=1}^T \sum_{i=1}^m p_i \cdot e_{i,t}
> $$
> By the assumption of correct coverage, for every $t$, there exists at least one $i$ such that $e_{i,t} = 0$, and since $p_i > 0$, we have:
> $$
> \sum_{i=1}^m p_i \cdot e_{i,t} < 1
> $$
> Now consider any model $M_j$. If there exists any $t$ such that $e_{j,t} = 1$ but $\sum_{i=1}^m p_i \cdot e_{i,t} < 1$, then:
> $$
> \sum_{t=1}^T \sum_{i=1}^m p_i \cdot e_{i,t} < \sum_{t=1}^T e_{j,t}
> \Rightarrow \mathbb{E}[E_{\mathrm{ens}}] < E_j
> $$
> Since such $t$ must exist unless $M_j$ is correct at all steps (which contradicts the existence of error in practice), the inequality is strict.
>
> We now proved that a randomized ensemble, formed by sampling agents independently at each step, achieves a strictly lower expected total error than any single fixed agent. This result provides a theoretical justification for leveraging model heterogeneity and stochastic selection to enhance robustness and accuracy in sequential reasoning tasks.
>
> We appreciate your valuable suggestion and will address it in a subsequent version of the paper.
>
> ### Weakness 2 & Question 2: Missing reflection examples
>
> Reflection: The failure in the previous trial occurred because the action to directly click "Buy Now" was executed without first ensuring that the selected item met all the specified criteria, such as being a long clip-in hair extension, natural looking, and priced under \$40.00. The initial action bypassed the necessary steps of verifying these details.
>
> Reflection: I appeared to have retrieved the correct information about The Oberoi Family and the location of it's head office, and provided a corresponding answer. However this answer does not exactly match the ground truth answer so I should try a different wording, such as Delhi.
>
> We appreciate your valuable suggestion and will address it in a subsequent version of the paper.
>
> ### Weakness 3 & Question 1: When SYMPHONY might fail?  Is it possible that bringing in more diverse models actually made things worse?
>
> Thank you for your insightful question. We fully acknowledge that while introducing diversity among models can enhance exploration, it also has the potential to introduce noise or inconsistency if not carefully controlled.
>
> This issue is directly examined in our Appendix G, where we perform targeted experiments designed to amplify disagreement and behavioral variance among agents through adversarial prompting and temperature scaling. In particular, the adversarial prompting strategy forces each agent to avoid replicating actions taken by its siblings, thereby maximizing diversity in candidate expansions. This results in a high frequency of fully unique child nodes (4-Unique expansions) at the same search layer. However, this artificially induced extreme heterogeneity leads to a noticeable decline in task performance, confirming that excessive diversity can indeed be detrimental.
>
> By contrast, our default SYMPHONY setup employs a more natural and balanced form of model heterogeneity by selecting large language models with differing architectures and pretraining data. These models produce outputs that are diverse yet semantically coherent without explicit adversarial constraints. Our empirical results in Section 4.5 demonstrate that the diversity brought by heterogeneity improves branch uniqueness and overall task accuracy, effectively balancing exploration with consistency.
>
> Furthermore, the multi-step Monte Carlo Tree Search process helps mitigate risks associated with individual model errors. Multiple candidate actions are generated at each expansion, and the Upper Confidence Bound (UCB)-based agent scheduler prevents any single model from dominating the search. Even when misleading or noisy actions arise, our Entropy-Modulated Confidence Scoring mechanism (Section 3.5) appropriately downweights low-confidence evaluations during value estimation.
>
> Together, this combination of structured search, adaptive agent selection, and uncertainty-aware scoring ensures that SYMPHONY maintains robustness against local noise or conflicting agent reflections, even under challenging or noisy conditions.
>
> In summary, while we do observe performance degradation under artificially maximized diversity, our standard heterogeneous agent configuration is deliberately designed to preserve coherence while leveraging complementary strengths. This careful balance allows SYMPHONY to improve performance reliably without suffering from the adverse effects of excessive model diversity.
>
>
> ### Question 3 and 4: Reflection mechanism
>
> Thank you for your insightful questions. The reflection mechanism is an important component of our system, enabling continual adaptation through natural language summaries without parameter updates, as detailed in Section 3.4.
>
> To clarify its usage frequency, reflections are triggered only when a search trajectory ends unsuccessfully. At such termination points, a UCB-selected agent generates a reflection summarizing the failure. These reflections are stored in a shared memory buffer accessible to all agents in future rollouts, allowing them to learn from diverse reasoning perspectives.
>
> Specifically:
>
> ● The reflection mechanism is not invoked at every search step but selectively activated only upon failed trajectory termination. Due to the MCTS design, which backpropagates low values and visit counts to discourage revisiting poor branches, the chance of encountering the same failure repeatedly is low. Thus, reflections act as lightweight cautionary signals for unexplored or uncertain branches rather than frequent interventions throughout the rollout.
>
> ● We maintain a fixed-size FIFO memory buffer that stores only the three most recent reflections. This small buffer aligns with MCTS behavior: failures are backpropagated to reduce revisit likelihood, making repeated errors unlikely. The reflections serve primarily as brief warnings for yet-unseen failure cases.
>
> ● Each reflection is a short natural language snippet held temporarily in runtime memory during a single task episode, keeping the memory footprint minimal. After task completion, reflections are discarded, ensuring resource efficiency and enabling efficient multi-agent sharing without redundant storage.
>
> ● Our relatively small rollout budget further limits failure trajectory occurrences, reducing the need for a larger reflection memory.
>
> While we have not extensively explored varying buffer sizes, prior work such as Reflexion (Shinn et al., NeurIPS 2023)  indicates that increasing reflections can boost performance but tends to plateau, with diminishing returns beyond a certain size. We appreciate this suggestion and consider it a valuable direction for future research.
>
> ### Question 5: Particular types of scenarios where SYMPHONY doesn’t work as well? For instance, in messier tasks where the signals are noisy or the goals aren’t well-defined, does it tend to struggle there?
>
> Thank you for this insightful question. You are absolutely right—SYMPHONY does face challenges in scenarios where task signals are noisy, goals are ambiguous, or the problem structure is particularly complex. These conditions make perfect performance difficult.
>
> For example, many tasks in the WebShop dataset involve such complexities. On this dataset, strong baselines like LATS achieve a success rate around 0.38, while SYMPHONY improves this to 0.72. Although we have made significant progress in addressing these challenges, the problem is not fully solved, and improving robustness in such noisy or ill-defined scenarios remains an important direction for future research.
>
> We sincerely appreciate your careful review. We hope our clarifications have resolved your concerns, and we would be happy to engage in further discussion if needed.
>
> ### References:
> Shinn, N., Cassano, F., Gopinath, A., Narasimhan, K., & Yao, S. (2023). Reflexion: Language agents with verbal reinforcement learning. Advances in Neural Information Processing Systems, 36, 8634–8652.

---

> > ### Comment · Reviewer_y451 · 2025-08-05
> > **Thank You.**
> >
> > I sincerely thank the authors for their detailed explanation. Most of my concerns have been addressed, and I will keep my original score.

---

> > > ### Author Response · Authors · 2025-08-05
> > >
> > > Thank you very much for your thoughtful comments and for taking the time to engage with our work. We truly appreciate your feedback, which has been highly constructive and will be fully incorporated into the final version.
> > >
> > > If any further questions arise or if there are additional aspects you’d like us to elaborate on, we would be more than happy to continue the discussion and share more details.
> > >
> > > Once again, we sincerely thank you for your time and consideration.

---

### Note · Authors · 2025-08-12

We thank the AC and reviewers for their constructive engagement. This work addresses a fundamental limitation: multi-path search algorithms with homogeneous strategies often collapse into a single path, narrowing exploration and increasing vulnerability to local optima and repeated failures.

We introduce the first heterogeneous model pool for LLM-based planning, where agents with distinct architectures, pretraining corpora, and inductive biases provide complementary reasoning behaviors. This diversity expands the search space and mitigates collapse. We validate its effectiveness through both mathematical theory and extensive experiments, showing that under mild conditions it achieves strictly lower expected trajectory error than any single constituent model.

The pool is integrated with three synergistic components within the MCTS search loop: UCB-based Agent Scheduling balances exploration and exploitation, Pool-wise Memory Sharing enables continual parameter-free adaptation via shared reflections, and Entropy-Modulated Node Evaluation refines value estimates using agent-level uncertainty.Together they sustain decision diversity, improve accuracy, and enhance robustness within MCTS.

Comprehensive experiments on HotpotQA, WebShop, MBPP-Python, and MBPP-Rust show SYMPHONY achieves the best task performance and lowest resource consumption among all baselines. In the high-capability setting (SYMPHONY-L) with large cloud-based API models, it attains the highest accuracy while reducing costs; in the lightweight setting (SYMPHONY-S) with only 7B and 8B models, it surpasses GPT-4 performance, demonstrating scalability and efficiency. Diversity Analysis shows path diversity strongly correlates with performance; Efficiency and Cost Analysis confirms minimal token and memory usage; Alternative Diversity Enhancements highlight the instability of other methods; and ablations verify each module’s contribution.

During rebuttal, we further strengthened our work: we provided more rigorous mathematical derivations and proofs, added a Single Strong Model Baseline confirming the effectiveness of other components, demonstrated compatibility with reasoning-oriented models, and reinforced our findings through Statistical Significance Analysis and concrete case studies.

In conclusion, SYMPHONY offers a theoretically grounded, empirically validated, and cost-efficient solution to a core bottleneck in LLM-based planning, combining conceptual novelty with practical superiority.

---

### Decision · Program_Chairs · 2025-09-17

**Decision:**

Accept (poster)

**Comment:**

SYMPHONY is a multi-agent framework for LLM-based planning that integrates different models within MCTS. Authors argue that while single-agent search often collapses into narrow, repetitive rollouts, coordinating multiple models with distinct inductive biases can expand the search space, improve exploration, and yield more robust plans.

There are three components—UCB-based agent scheduling, shared reflection memory, and entropy-modulated node evaluation. Approach is tested on benchmarks spanning reasoning (HotpotQA), decision-making (WebShop), and code generation (MBPP). Results show improvements over baselines, ToT, Reflexion, LATS, and MASTER, with both lightweight (open-source only) and large (API-based) configurations.

Reviewers agree that the problem is timely and well motivated, the paper is clearly presented, and the results are strong across tasks. They found the heterogeneity compelling: rather than ensembling similar models, the authors demonstrate how architectural and pretraining diversity introduces meaningful exploration patterns.

Reviewers also appreciated the ablations, cost–efficiency analysis, and the new experiments in rebuttal, including single-strong-model baselines, reasoning-model compatibility, and case studies illustrating how weaker models help escape local failure modes. The authors also provided a theoretical argument showing why randomized heterogeneous ensembles should lower expected trajectory error, further grounding the approach.

The main weaknesses were about novelty and framing: some felt the individual components (UCB, reflections, entropy scoring) are straightforward, and the contribution lies more in their integration than in algorithmic innovation. Concerns were also raised about missing discussion of failure cases, stability, and alignment with real-world agent settings beyond curated benchmarks. Still, the rebuttal addressed these points convincingly, and the combination of conceptual clarity, practical efficiency, and empirical strength makes this a solid contribution. I recommend to accept as poster, not a spotlight, since the methodological novelty is moderate and rests more on system design than on a fundamental algorithmic advance.